# TGB 2.0: A Benchmark for Learning on Temporal Knowledge Graphs and Heterogeneous Graphs

**Julia Gastinger**[1,2,6*]   **Shenyang Huang**[1,4*]   **Mikhail Galkin**[3]   **Erfan Loghmani**[8]
**Ali Parviz**[1,9]   **Farimah Poursafaei**[1,4]   **Jacob Danovitch**[1,4]
**Emanuele Rossi**[5]   **Ioannis Koutis**[9]   **Heiner Stuckenschmidt**[2]
**Reihaneh Rabbany**[1,4,7]   **Guillaume Rabusseau**[1,6,7]
[1]Mila - Quebec AI Institute, [2]Mannheim University, [3]Intel AI Lab,
[4]School of Computer Science, McGill University, [5]Imperial College London,
[6]DIRO, Université de Montréal, [7]CIFAR AI Chair,
[8]University of Washington [9]New Jersey Institute of Technology

## Abstract

Multi-relational temporal graphs are powerful tools for modeling real-world data, capturing the evolving and interconnected nature of entities over time. Recently, many novel models are proposed for ML on such graphs intensifying the need for robust evaluation and standardized benchmark datasets. However, the availability of such resources remains scarce and evaluation faces added complexity due to reproducibility issues in experimental protocols. To address these challenges, we introduce Temporal Graph Benchmark 2.0 (TGB 2.0), a novel benchmarking framework tailored for evaluating methods for predicting future links on Temporal Knowledge Graphs and Temporal Heterogeneous Graphs with a focus on large-scale datasets, extending the Temporal Graph Benchmark. TGB 2.0 facilitates comprehensive evaluations by presenting eight novel datasets spanning five domains with up to 53 million edges. TGB 2.0 datasets are significantly larger than existing datasets in terms of number of nodes, edges, or timestamps. In addition, TGB 2.0 provides a reproducible and realistic evaluation pipeline for multi-relational temporal graphs. Through extensive experimentation, we observe that 1) leveraging edge-type information is crucial to obtain high performance, 2) simple heuristic baselines are often competitive with more complex methods, 3) most methods fail to run on our largest datasets, highlighting the need for research on more scalable methods.

## 1 Introduction

Learning from graph-structured data has become ubiquitous in many applications such as recommendation systems [35, 70], knowledge base completion [55, 46] and molecular learning [58, 3]. Relational data often evolves over time and can contain multiple types of relations. These complex interactions and temporal dependencies can be captured by *multi-relational* temporal graphs. In recent years, various approaches have emerged to predict future links in such graphs, notably for prediction on Temporal Knowledge Graphs (TKGs) [41, 47] and Temporal Heterogeneous Graphs (THGs) [38, 30]. These approaches capture the rich information from multi-relational data, forming distinct lines of research from those of single-relational temporal graphs [61, 50]. However, benchmarking on multi-relational temporal graphs faces two main challenges: *inconsistent evaluation* and *limited dataset size*.

---

*Equal contributions

38th Conference on Neural Information Processing Systems (NeurIPS 2024) Track on Datasets and Benchmarks.

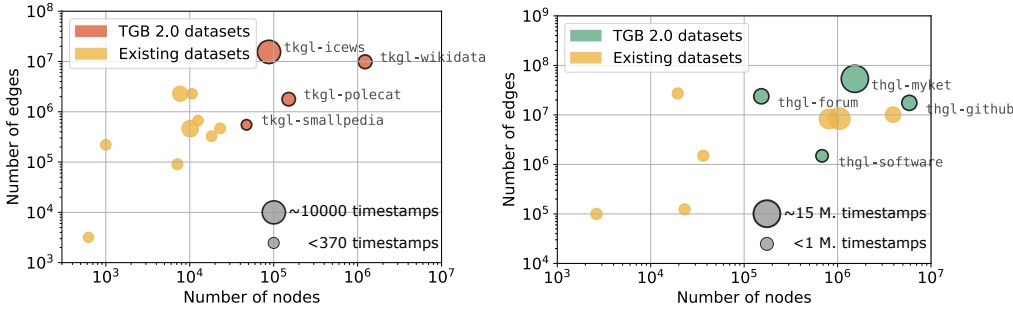

(a) Novel Temporal Knowledge Graphs       (b) Novel Temporal Heterogeneous Graphs

Figure 1: Existing benchmark datasets (yellow) vs. novel datasets in TGB 2.0 for TKG (a) marked in orange and THG (b) marked in green. Circle sizes correspond to the number of timestamps. TGB 2.0 datasets are significantly larger than existing datasets in number of nodes, edges and timestamps.

**Inconsistent Evaluation.** Evaluation for multi-relational temporal graphs faces significant challenges. Recently, it was shown that existing evaluation for TKGs has inconsistencies in a) the evaluation metrics, b) mixing multi-step and single-step prediction settings and c) using different versions of the same dataset [16]. Similar inconsistencies have been observed in related areas such as link prediction on static knowledge graphs [67, 60], node and graph classification on static graphs [64, 13], and temporal graph link prediction [28]. In addition, for link prediction on THGs, existing evaluation often includes a single random negative per positive edge [38, 72], leading to over-optimistic performances, inconsistent evaluation, and reducing performance differentiation between methods [57].

**Limited Dataset Size.** Existing evaluations are conducted on predominantly small-scale datasets. For example, commonly used TKG and THG datasets consist of less than two million edges and one million nodes [38, 30, 41, 39]. However, real world networks typically contain tens of millions of nodes and edges thus existing datasets rarely reflect the true scale of datasets in practice. In addition, significant efforts were made to design scalable graph learning methods for real applications which requires the availability of large scale datasets [27, 26, 28]. These challenges hinder meaningful comparisons between methods and the accurate assessment of progress, and hamper advancements in the field. Therefore, there is an urgent need for a public and standardized benchmark to facilitate proper and fair comparison between methods, accelerating research for multi-relational temporal graphs.

To address the aforementioned challenges, we present TGB 2.0, a novel benchmark designed for future link prediction on multi-relational temporal graphs. Building upon the foundations of the Temporal Graph Benchmark (TGB) [29] where only *single-relation temporal graphs* are included, TGB 2.0 introduces *multi-relational temporal graph* datasets. TGB 2.0 adds four novel TKG datasets and four novel THG datasets of varying scale, spanning five domains. Figure 1 shows the difference in scale of the novel datasets in TGB 2.0 when compared to existing ones. Figure 1a shows that TGB 2.0 TKG datasets (marked in orange) are orders of magnitude larger than existing ones in terms of number of nodes, edges and timestamps. Figure 1b shows that TGB 2.0 THG datasets (marked in green) are significantly larger than existing datasets, with `thgl-myket` dataset quintupling the number of edges and timestamps, while `thgl-github` has the most number of nodes to date. Additionally, TGB 2.0 provides an automated evaluation pipeline for reproducible and realistic evaluation on multi-relational temporal graphs. In TGB 2.0, the dynamic link property prediction task is treated as a ranking problem where multiple negative edges are ranked against the positive edge. For large datasets, we sample negative edges based on the edge type of the query, and thus closely approximate the complete evaluation where all negative edges are used. Overall, TGB 2.0 presents a benchmark for realistic, challenging, and reproducible evaluation on *multi-relational temporal graphs* while providing an automated pipeline for dataset downloading, processing, evaluation as well as a public leaderboard. TGB 2.0 has the following main contributions:

- **Large and diverse datasets for multi-relational graphs.** We present four novel TKGs that are orders of magnitude larger than existing ones and four novel THGs that are significantly larger in number of nodes, edges and timestamps when compared with current ones.

- **Realistic and reproducible evaluation.** We provide an evaluation pipeline for multi-relational temporal graphs, which automates the dataset downloading, processing and benchmarking process

for seamless reproducibility. TGB 2.0 evaluation uses the ranking metric MRR and samples challenging negative edges based on the edge type information, hence providing realistic evaluation.

- **Experimental insights.** The main insight from our experiments is that for both THGs and TKGs, all methods (apart from heuristics) fail to scale to our largest datasets, highlighting the need for more research on scalable methods. Surprisingly, the heuristic baselines perform competitively with more sophisticated methods. On THG datasets, we observe that methods that leverage the edge type and node type information achieve strong performance. Finally, across TKG datasets, we observe a strong correlation between the recurrency degree of a given relation type and the performance of methods, suggesting large room for improvement on low recurrency relations.

**Reproducibility.** TGB 2.0 code and datasets are publicly available (see Appendix C for download links) and theTGB 2.0 website provides detailed documentation.

## 2   Preliminaries

**Temporal Knowledge Graphs:** A *Temporal Knowledge Graph* (TKG) $G_K$ is a set of quadruples $(s, r, o, t)$ with subject and object entities $s, o \in V$ (the set of entities), relation $r \in R$ (the set of possible relations), and timestamp $t$. The semantic meaning of a quadruple $(s, r, o, t)$ is that $s$ is in relation $r$ to $o$ at time $t$. We also refer to quadruples as temporal triples, or simply as edges.

**Temporal Heterogeneous Graphs:** A *Temporal Heterogeneous Graph* (THG) $G_H$ is similarly a set of quadruples, $(s, r, o, t) \in E$ where $s, o \in V$ are entities, $r$ is the relation and $t$ is a timestamp, *along with a node type function* $\phi : V \to A$. THG are equivalent to TKG with the addition that each node is assigned a fixed type (consistent over time) by the node type function.

**Temporal Graph Forecasting (Extrapolation):** Given a Temporal multi-relational Graph $G_K$ or $G_H$, Temporal Graph Forecasting or Extrapolation is the task of predicting edges for *future* timesteps $t^+$. Akin to (static) multi-relational graph completion, temporal multi-relational graph forecasting is approached as a ranking task [21]. For a given query, e.g. $(s, r, ?, t^+)$, methods rank all entities in $V$ as possible objects using a scoring function, assigning plausibility scores to each quadruple. In TGB 2.0, we focus on the temporal graph forecasting task.

**Time Representations.** There are two approaches for representing time in temporal graphs: (a) *discrete*, where graphs are modeled as snapshots $G_t$ containing all edges appearing at time $t$, and (b) graphs that can be conceptualized as a series of edges arriving at *continuous* timestamps. In practice, TKGs are often represented as snapshots and TKG methods are tailored for discrete time representations [41, 39, 47, 66]. This choice is driven by the discrete nature of the data sources and the suitability of snapshot-based representations for downstream tasks. Therefore, we represent TKGs as snapshots in TGB 2.0. In comparison, THG data sources are often continuous in nature (recorded in second-wise interactions), while both discrete and continuous approaches for THGs are developed [38, 10, 72]. It is often argued that continuous representations preserve more information and can be converted to discrete while the reverse is not true [33]. Therefore, the THG datasets are represented in the continuous format.

## 3   Related Work

**TKG Methods.**

Most TKG forecasting methods utilize a discrete time representation, except for [24]. Some methods integrate the message-passing paradigm from static graphs [63, 53] with sequential techniques [32, 41, 22, 23, 39, 45]. Other approaches combine Reinforcement Learning with temporal reasoning for future link prediction [40, 66]. Rule-based methods [47, 34, 52, 48, 44] employ strategies to learn temporal logic rules while others [76, 71, 75] combines a blend of different methodologies. More details are in Appendix H.1. For more comprehensive discussions on methods and applications of TKG, we refer to the surveys [8], [9], [43], and [69]. For TKG forecasting, common benchmark datasets include YAGO [51], WIKI [36, 31], GDELT [37] and the Integrated Crisis Early Warning System (ICEWS) dataset [6]. However, these datasets are orders of magnitude smaller than our TKG datasets in number of nodes, edges and timestamps. In tkgl-icews, we include the full ICEWS dataset [6] spanning 28 years when comparing to prior versions containing only one or a

few years [14, 31, 11]. Similarly, our `tkgl-wikidata` dataset is orders of magnitude larger than the existing WIKI dataset [15, 42] in size of nodes, edges and timestamps.

**THG Methods.** THG methods can be categorized based on their time representation: discrete-time methods and continuous-time methods. Examples of continous time methods include HTGN-BTW [74] and STHN [38]. HTGN-BTW [74] enables TGN [61] to accommodate heterogeneous node and edge types. STHN [38] utilizes a link encoder and patching techniques to incorporate edge type and time information respectively. Discrete-time methods include random walk based methods [5, 73] and message-passing based methods[72, 10]. However, it is difficult to adapt discrete-time methods for continuous-time datasets. More details are provided in Appendix H.2. Common THG datasets such as MathOverflow [56], Netflix [4] and Movielens [25] are small and only contain a few million edges [38]. Large datasets such as Dataset A and B from the WSDM 2022 Challenge and the TRACE and THEIA datasets [59] are only evaluated with one negative sample per positive edge, which is shown to lead to over-optimistic and insufficient evaluation [28, 57]. Here, we introduce the large `thgl-myket` dataset with 53 million edges and 14 million timestamps.

**Graph Learning Benchmarks.** The Open Graph Benchmark (OGB) [27] and the OGB large scale challenge [26] are popular benchmarks accelerating progress on static graphs. Recently, the Temporal Graph Benchmark was introduced for temporal graph learning, consisting of large datasets for single-relation temporal graphs [28]. In this work, we introduce novel TKG and THG datasets, incorporating multi-relational temporal graphs into TGB. Recently, detailed performance comparison for deep learning methods on dynamic graphs are conducted in [19], however multi-relational temporal graph datasets were not included in the comparison. While efforts like [16] have highlighted evaluation inconsistencies in TKG, their study focuses on existing smaller-scale datasets where no novel evaluation framework were proposed. Moreover, recent findings by [15] reveal that a simple heuristic baseline outperforms existing methods on some datasets, thus underscores the necessity for comparison with baselines. In this work, TGB 2.0 includes four novel TKG and four novel THG datasets as well as a standardized and reproducible evaluation pipeline.

## 4 Datasets

TGB 2.0 introduces eight novel datasets from five distinct domains consisting of four TKGs and four THGs. We split all datasets chronologically into training, validation, and test sets, respectively containing $70\%$, $15\%$, and $15\%$ of all edges in line with existing studies [28, 61, 50] and ensure that edges of a timestamp can only exist in either train or validation or test set[2]. We present the dataset licenses and download links in Appendix D. The datasets will be permanently maintained via Digital Research Alliance of Canada (funded by the Government of Canada).

**Dataset Details.** Here we describe each TGB 2.0 dataset in detail. Temporal Knowledge Graph datasets start with the prefix `tkgl-` while Temporal Heterogeneous Graph datasets start with `thgl-`. More details on the dataset collection process are in Appendix E.

`tkgl-smallpedia`. This TKG dataset is constructed from the Wikidata Knowledge Graph [68] where it contains facts paired with Wikipedia pages. Each fact connects two entities via an explicit relation (edge type). This dataset contains Wikidata entities with IDs smaller than 1 million. The temporal relations either describe point in time relations (event-based) or relations with duration (fact-based). We also provide static relations from the same set of Wikidata pages which include 978,315 edges that can be used to enhance model performance. The task is to predict future facts.

`tkgl-polecat`. This TKG dataset is based on the POLitical Event Classification, Attributes, and Types (POLECAT) dataset [62] which records coded interactions between socio-political actors of both cooperative or hostile actions. POLECAT utilizes the PLOVER ontology [20] to analyze new stories in seven languages across the globe to generate time-stamped, geolocated events. These events are processed automatically via NLP tools and transformer-based neural networks. This dataset records events from January 2018 to December 2022. The task is to predict future political events between political actors.

`tkgl-icews`. This TKG dataset is extracted from the ICEWS Coded Event Data [7, 65] which spans a time frame from 1995 to 2022. The dataset records political events between actors. It is classified

---

[2]We detail the exact number of timestamps and edges for each subset in Appendix G.8

Table 1: Dataset information including common statistics and the proportion of Inductive Test nodes (Induct. Test Nodes), the Direct Recurrency Degree (DRec), the Recurrency Degree (Rec), the Consecutiveness Value (Con), as well as the mean number of edges and nodes per timestep (Mean Edges/Ts. and Mean Nodes/Ts.)

| | Temporal Knowledge Graphs (`tkgl-`) | | | | Temporal Heterogeneous Graphs (`thgl-`) | | | |
|---|---|---|---|---|---|---|---|---|
| Dataset | `smallpedia` | `polecat` | `icews` | `wikidata` | `software` | `forum` | `github` | `myket` |
| Domain | knowledge | political | political | knowledge | software | social. | software | interac. |
| # Quadruples | 550,376 | 1,779,610 | 15,513,446 | 9,856,203 | 1,489,806 | 23,757,707 | 17,499,577 | 53,632,788 |
| # Nodes | 47,433 | 150,931 | 87,856 | 1,226,440 | 681,927 | 152,816 | 5,856,765 | 1,530,835 |
| # Edge Types | 283 | 16 | 391 | 596 | 14 | 2 | 14 | 2 |
| # Node Types | - | - | - | - | 4 | 2 | 4 | 2 |
| # Timesteps | 125 | 1,826 | 10,224 | 2,025 | 689,549 | 2,558,457 | 2,510,415 | 14,828,090 |
| Granularity | year | day | day | year | second | second | second | second |
| Induct. Test Nodes | 0.26 | 0.12 | 0.05 | 0.34 | 0.13 | 0.02 | 0.14 | 0.01 |
| DRec | 0.71 | 0.07 | 0.11 | 0.61 | 0.00 | 0.00 | 0.00 | 0.00 |
| Rec | 0.72 | 0.43 | 0.63 | 0.61 | 0.10 | 0.63 | 0.01 | 0.37 |
| Con | 5.82 | 1.07 | 1.14 | 5.05 | 1.00 | 1.00 | 1.00 | 1.00 |
| Mean Edges/Ts. | 4,403.01 | 974.59 | 1,516.91 | 4,867.26 | 0.56 | 8.87 | 6.54 | 3.15 |
| Mean Nodes/Ts. | 5,289.16 | 550.60 | 1,008.65 | 5,772.16 | 0.86 | 12.96 | 9.77 | 6.24 |

based on the CAMEO taxonomy of events [18] which is optimized for the study of mediation and contains a number of tertiary sub-categories specific to mediation. When compared to PLOVER ontology in `tkgl-polecat`, the CAMEO codes have more event types (391 compared to 16). The task is to predict future interactions between political actors.

`tkgl-wikidata`. This TKG dataset is extracted from the Wikidata KG [68] and constitutes a superset of `tkgl-smallpedia`. The temporal relations are properties between Wikidata entities. `tkgl-wikidata` is extracted from wikidata pages with IDs in the first 32 million. We also provide static relations from the same set of Wiki pages containing 71,900,685 edges. The task is to forecast future properties between wiki entities.

`thgl-software`. This THG dataset is based on Github data collected by GH Arxiv. Only nodes with at least 10 edges were kept in the graph, thus resulting in 14 types of relations and 4 node types (similar relations to [1]). The dataset spans January 2024. The task is to predict the next activity of a given entity, e.g., which pull request the user will close at a given time.

`thgl-forum`. This THG dataset is based on the user and subreddit interaction network on Reddit [54]. The node types encode users or subreddits, the edge relations are "user reply to user" and "user -post" in subreddits. The dataset contains interactions from January 2014. The task is to predict which user or subreddit a user will interact with at a given time.

`thgl-myket`. This THG dataset is based on the Myket Android App market. Each edge documents the user installation or update interaction within the Myket market. The data spans six months and two weeks and when compared to an existing smaller version [49], this dataset contains the full data without downsampling. Overall, the dataset includes information on 206,939 applications and over 1.3 million anonymized users from June 2020 to January 2021.

`thgl-github`. This THG dataset is based on Github data collected from the GH Arxiv. This is a large dataset from a different period from `thgl-software`. We extract user, pull request, issue and repository nodes and track 14 edge types. The nodes with two or fewer edges are filtered out. The dataset contains the network as of March 2024. The task is to predict the next activity of an entity.

**Varying Scale.** Table 1 shows the detailed characteristics of all datasets, such as the number of quadruples and nodes. TGB 2.0 datasets vary significantly in scale for number of nodes, edges, and time steps. We observe an increase in runtime and memory requirements from `tkgl-smallpedia` to `tkgl-polecat` to `tkgl-icews` and `tkgl-wikidata`. In practice, these requirements depend on the combination of number of nodes, edges and time steps. To account for such benchmarking requirements, we categorize the datasets into small, medium and large datasets. Small datasets are suitable for prototyping methods, while medium and large datasets test method performance at increasingly large scales.

Table 1 reports dataset statistics: the *Proportion of Inductive Test Nodes (Induct. Test Nodes)* is the proportion of nodes in the test set that have not been seen during training. The *Recurrency Degree*

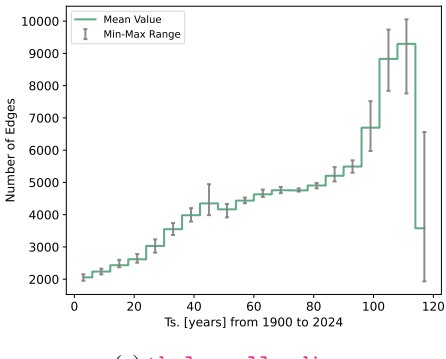

(a) `tkgl-smallpedia`

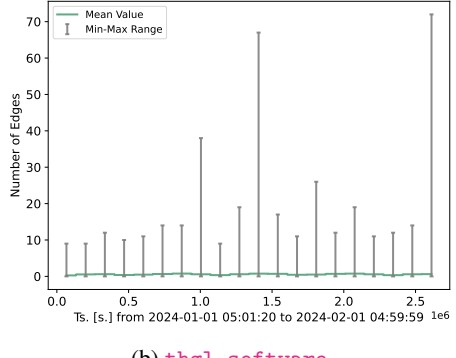

(b) `thgl-software`

Figure 2: Number of edges over time

*(Rec)*, which is defined as the fraction of test temporal triples $(s, r, o, t^+)$ for which there exists a $k < t^+$ such that $(s, r, o, k) \in G$. The *Direct Recurrency Degree (DRec)* which is the fraction of temporal triples $(s, r, o, t^+)$ for which it holds that $(s, r, o, t^+ - 1) \in G$ [15]. Also, we represent a novel metric called *Consecutiveness Value (Con)*, which quantifies if a given temporal triples repeats at consecutive timestamps by averaging the maximum number of consecutive timesteps during which a triple holds true across all triples in the dataset. Intuitively, fact-based relations which are true across multiple consecutive time steps will result in a higher *Consecutiveness Value*.

**Diverse Statistics.** TGB 2.0 datasets exhibits diverse dataset statistics. For example, `tkgl-wikidata`, `tkgl-smallpedia`, `tkgl-polecat` and `thgl-software` all have more than 10% test nodes that are inductive (i.e. nodes unseen in the training set), thus testing the inductive capability of methods. Variations in the recurrence of relations are evident with `tkgl-smallpedia` and `tkgl-wikidata` showing higher Recurrency Degrees compared to others. The DRec highlights the disparities between THG and TKG datasets, where the finer, second-wise time granularity of THG leads to a DRec of 0 implying no repetition of facts across subsequent time steps. On TKG datasets, the high Consecutiveness Value for `tkgl-smallpedia` and `tkgl-wikidata` exhibit a prevalence of long-lasting facts, contrasting with `tkgl-icews` and `tkgl-polecat` which documents political events. In comparison, THG datasets describe one-time events, thus displaying lower Con values.

Figure 2 shows the number of edges per timestamp for `tkgl-smallpedia` and `thgl-software`, reported over twenty bins with bars showing the min/max in each bin. Similar figures for other datasets are given in Appendix F. Figure 2 underscores distinctions between datasets, particularly in terms of time granularity and trend patterns. TKG datasets demonstrate a coarser time granularity leading to a significantly higher edge count per timestep compared to THG datasets. `thgl-software` exhibits relatively constant number of edges over time (with peaks at specific time points). In comparison, `tkgl-smallpedia` exhibit significant growth in edge count closer to the end. This is because `tkgl-smallpedia` starts from 1900 and ends at 2024, as time gets closer to current era, the amount of digitized and documented information increases significantly. The reduced number of edges in the final bin is due to the fact that the knowledge from 2024 remains incomplete as of this writing.

Figure 3 illustrates the distribution of the ten most prominent relations in `tkgl-smallpedia` and `thgl-software`. More Figures are in Appendix F. There are highly frequent relation types in `tkgl-smallpedia` such as member of sports team which occupies 28% of all edges; the portion of edges quickly reduces for other relations. In `thgl-software`, there is a relatively even split in the portion of edges for the most prominent relations with the top seven relations each occupying more than 10% of edges. These figure show the diversity of relations and their distributions in TGB 2.0.

## 5 Experiments

**Evaluation Protocol.** In TGB 2.0, we focus on the *dynamic link property prediction task* where the goal is to predict the property (often existence) of a link between a pair of nodes in a future timestamp. Here, we treat the link prediction task as ranking problem similar to [27, 28, 16]. The model is required to assign the true edge with the highest probability from multiple negative edges (also referred to as corrupted triples in the TKG literature). The evaluation metric is the time-aware filtered Mean Reciprocal Rank (MRR) following [16, 28]. The MRR computes the average of the

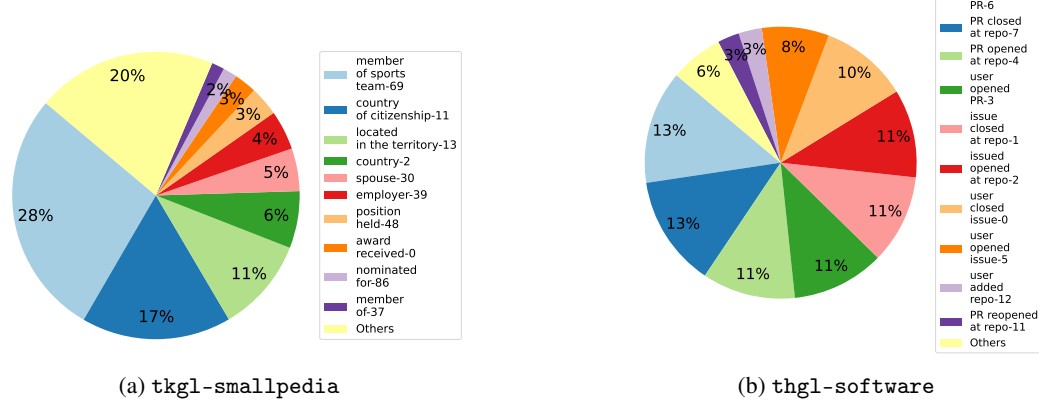

(a) `tkgl-smallpedia`                     (b) `thgl-software`

Figure 3: Most frequent relation types for `tkgl-smallpedia` and `thgl-software` datasets. *Others* refers to all remaining relations not shown here.

reciprocals of the ranks of the first relevant item in a list of results. The time-aware filtered MRR removes any edge that are known to be true at the same time as the true edge (i.e. temporal conflicts) from the list of possible destinations. For THG datasets, we predict the tails of queries $(s, r, ?, t^+)$, as in [38, 61]. Following the practice in TKG literature [16], we predict entities in both directions for TKG datasets, namely both $(s, r, ?, t^+)$ and $(?, r, o, t^+)$, achieved by introducing inverse relations where the head and tail of an existing relation is inverted. Due to the large size of TGB 2.0 datasets, we select the number of negative edges $q$ for each dataset considering the trade-off between the evaluation completeness and the test inference time. Therefore, we utilize two negative sampling strategies for evaluation: *1-vs-all* and *1-vs-q*. For both strategies, the temporal conflicts are removed for correctness. All negative samples are then pre-generated to ensure reproducible evaluation. Lastly, any methods that uses more than 40 GB GPU memory or runs for more than a week are considered as Out Of Memory (OOM) or Out Of Time (OOT), respectively.

*1-vs-all*. For datasets with a small number of nodes, it is possible to evaluate with all the possible destinations, thus achieving a comprehensive evaluation. In TGB 2.0, we use *1-vs-all* strategy for `tkgl-smallpedia`, `tkgl-polecat` and `tkgl-icews` due to their smaller node size (see Table 1).

*1-vs-q*. For datasets with a large number of nodes, sampling $q$ negative edges is required to achieve a practically feasible inference time. We find that randomly sampling the negative edges, omitting the edge types, results in over-optimistic MRRs, making the prediction easy. We thus propose to incorporate the edge type information into the negative sampling process for more robust evaluation. For the large TKG dataset `tkgl-wikidata`, we first identify possible tails for each edge type throughout the dataset and then sample the negatives based on the edge type of the query. If there are not enough tails in a given edge type, we then randomly sample the remaining ones. For all THG datasets, we sample all destination nodes with the same node type as the true destination node, thus considering the tail node type associated with a given edge type. We conduct an ablation study to show the effectiveness of our sampling strategy in Appendix G.5 on the `tkgl-smallpedia` dataset. We find that our sampling results in closer MRR to that of the *1-vs-all* than random sampling.

## 5.1 Temporal Knowledge Graph Experiments

For TKG experiments, we include RE-GCN [41], TLogic [47], CEN [39] as well as two deterministic heuristic baselines: the Recurrency Baseline (RecB) [15] and EdgeBank [57]. For RecB, we report two versions where applicable: RecB$_{default}$ which uses default values for its two parameters, and RecB$_{train}$ which selects the optimal values for these based on performance on the validation set. For EdgeBank, we report two versions following [57], EdgeBank$_{tw}$, which accounts for information from a fixed past time window, and EdgeBank$_{\infty}$, which uses information from all past temporal triples. Method details and compute resources are in Appendix H.1 and Appendix G.1, respectively.

We report the average performance and standard deviation across 5 runs for each method in Table 2. The runtimes and GPU usage results are in Appendix G.3 and G.2. In particular, several methods encountered out of memory or out of time errors on some datasets. The results reveals

Table 2: **MRR** results for *Temporal Knowledge Graph Link Prediction* task. We report the average and standard deviation across 5 runs. First place is **bolded**, second place is underlined.

| Method | tkgl-smallpedia | | tkgl-polecat | | tkgl-icews | | tkgl-wikidata | |
|---|---|---|---|---|---|---|---|---|
| | Validation | Test | Validation | Test | Validation | Test | Validation | Test |
| EdgeBank$_{tw}$ [57] | 0.457 | 0.353 | 0.058 | 0.056 | 0.020 | 0.020 | 0.633 | 0.535 |
| EdgeBank$_\infty$ [57] | 0.401 | 0.333 | 0.048 | 0.045 | 0.008 | 0.009 | 0.632 | 0.535 |
| RecB$_{train}$ [15] | 0.639 | 0.605 | 0.203 | 0.198 | 0.270 | **0.211** | OOT | OOT |
| RecB$_{default}$ [15] | 0.542 | 0.486 | 0.170 | 0.167 | 0.264 | 0.206 | OOT | OOT |
| RE-GCN [41] | 0.631±0.001 | 0.594±0.001 | 0.191±0.003 | 0.175±0.002 | 0.232±0.003 | 0.182±0.003 | OOM | OOM |
| CEN [39] | **0.646**±0.001 | **0.612**±0.001 | 0.204±0.002 | 0.184±0.002 | 0.244±0.002 | 0.187±0.003 | OOM | OOM |
| TLogic [47] | 0.631±0.000 | 0.595±0.001 | **0.236**±0.002 | **0.228**±0.001 | **0.287**±0.001 | 0.186±0.001 | OOT | OOT |

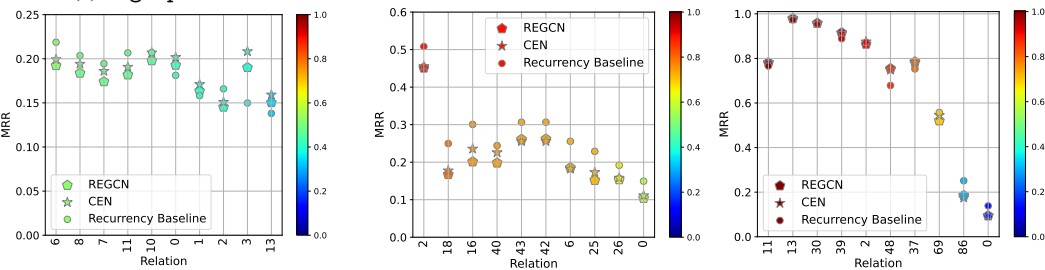

Figure 4: MRR per relation for the 10 highest occuring relations for three TKG datasets for RE-GCN, CEN and RecB$_{train}$. The color indicates the Recurrency Degree value for relation type. The relations for each dataset are ordered by decreasing Recurrency Degrees.

that no single method exhibits superior performance across all four datasets. Surprisingly, the RecB heuristic performs competitively across most datasets while being among the best performing on `tkgl-smallpedia` and `tkgl-icews`, underscoring the importance of including simple baselines in comparison and suggesting potential areas for improvement in other methods. The Edgebank heuristic, originally designed for homogenous temporal graphs, exhibits low performance, highlighting the importance of utilizing the rich multi-relational information for TKG learning. On the large `tkgl-wikidata` dataset, however, Edgebank is the only method that can scale to such size, likely due to the fact that it omits edge type information. This highlights the need for scalable methods. On another note, methods achieve higher MRRs on datasets characterized by high Recurrency Degrees and Consecutiveness values (`tkgl-smallpedia`, `tkgl-wikidata`), despite the presence of a considerable number of inductive nodes in these datasets.

**Per-relation Analysis**. Figure 4 illustrates the performance per relation of selected methods across three datasets [3]. For each dataset, we show the ten most frequent relations, ordered by decreasing Recurrency Degree with the color reflecting the Recurrency Degree of each relation. Note that the y-axis scale varies across datasets. We observe distinct patterns in relation-specific performance across datasets: while results on `tkgl-polecat` exhibit consistent performance across relations, suggesting a relative homogeneity, results on the `tkgl-smallpedia` dataset shows significant variance, indicating a higher degree of variation among relations. Interestingly, there is strong correlation between the Recurrency Degree and performance, most evident within the `tkgl-smallpedia` dataset.

## 5.2 Temporal Heterogenous Graph Experiments

For THG experiments, we include TGN [61] (with and without edge type information), STHN [38], RecB [15], and EdgeBank [57]. Table 3 reports the average performance and standard deviation across 5 runs for each method. Scalability is a significant challenge for THG methods on large datasets such as `thgl-forum` and `thgl-myket`, more details can be found in Appendix G. Most methods either are out of memory or out of time for these datasets. STHN achieves the highest performance on `thgl-software` dataset showing methods designed for THG can achieve significant performance gain. However, STHN is the least scalable, requiring 185 GB of memory for `thgl-software` to compute subgraphs and unable to scale to other datasets. The widely-used TGN model [61] for single-relation temporal graph learning is also adapted here, with a modification to incorporate the edge type information as edge feature. We observe significant improvement when TGN utilizes edge

---

[3]The relation description can be found based on their relation ID in Figure 7 in Appendix.

Table 3: **MRR** results for *Temporal Heterogeneous Graph Link Prediction* task. We report the average and standard deviation across 5 runs. First place is **bolded**, second place is underlined.

| Method | thgl-software | | thgl-forum | | thgl-github | | thgl-myket | |
|---|---|---|---|---|---|---|---|---|
| | Validation | Test | Validation | Test | Validation | Test | Validation | Test |
| EdgeBank$_{tw}$ [57] | 0.279 | 0.288 | 0.534 | 0.534 | 0.355 | 0.374 | 0.248 | 0.245 |
| EdgeBank$_\infty$ [57] | 0.399 | 0.449 | 0.612 | 0.617 | 0.403 | 0.413 | 0.430 | 0.456 |
| RecB$_{default}$ [15] | 0.106 | 0.099 | 0.552 | 0.561 | OOT | OOT | OOT | OOT |
| TGN [61] | 0.299$_{\pm 0.012}$ | 0.324$_{\pm 0.017}$ | 0.598$_{\pm 0.086}$ | 0.649$_{\pm 0.097}$ | OOM | OOM | OOM | OOM |
| TGN$_{edge-type}$ | 0.376$_{\pm 0.010}$ | 0.424$_{\pm 0.013}$ | **0.767**$_{\pm 0.005}$ | **0.729**$_{\pm 0.009}$ | OOM | OOM | OOM | OOM |
| STHN [38] | **0.764**$_{\pm 0.025}$ | **0.731**$_{\pm 0.005}$ | OOM | OOM | OOM | OOM | OOM | OOM |

type data, thus showing the potential to leverage the multi-relational information in THGs. Lastly, EdgeBank achieves competitive performance with that of TGN while being scalable to large datasets. Thus, it is important to evaluate against simple baselines to understand method performances.

## 6 Conclusion

In this work, we introduce TGB 2.0, a novel benchmark for reproducible, realistic, and robust evaluation on multi-relational temporal graphs that is building on the Temporal Graph Benchmark (TGB). We present four new TKG and four new THG datasets which introduce multi-relation datasets in TGB. TGB 2.0 datasets are significantly larger than existing ones while being diverse in statistics and dataset domains. TGB 2.0 focuses on the dynamic link property prediction task and provides an automated pipeline for dataset downloading, processing, method evaluation, and a public leaderboard to track progress. From our experiments, we find that both TKG and THG methods struggle to tackle large scale datasets in TGB 2.0, often resulting in overly long runtime or exceeding the memory limit. Therefore, scalability is an important future direction. Another observation is that heuristic methods achieve competitive results on TKG and THG datasets. This highlights the importance of the inclusion of simple baselines and underlines the room for improvement for current methods.

**Limitations** This work exclusively considers the continuous-time setting for THG datasets. Depending on the application, either the continuous-time or discrete-time setting may be more appropriate. However, the continuous-time setting is often regarded as the more general framework. However, many THG methods are designed for discrete settings. Thus, in future work, discretized versions of the data sets could be added for comparative analysis between discrete methods. Additionally, the TGB 2.0 dataset collection currently includes datasets from only five distinct domains. Notably, domains such as biological networks and citation networks are not represented. To address this limitation, we plan to expand the dataset collection by incorporating additional datasets based on community feedback, thereby enhancing the diversity and comprehensiveness of the dataset repository. As temporal graph data often records interactions between individuals or other sensitive information, data privacy is a potential concern. Sensitive information may be stored as node level or link level attributes thus proper anonymization of data should always be a top priority. In this work, we anonymize user information where appropriate to prevent the leakage of identifiable information.

## Acknowledgment

We express our gratitude to Myket Corporation for generously providing the anonymous interaction data utilized in this research. We are thankful to Dr. Vahid Rahimian for agreeing to collaborate and for their support throughout the data-sharing process and to Ms. Zahra Eskandari for her diligent efforts in collecting and arranging the data. Additionally, we thank MohammadAmin Fazli for his continuous support and help in facilitating collaboration and data sharing. We thank Weihua Hu, Matthias Fey, Jure Leskovec and Michael Bronstein from the TGB team for their support and discussion. We thank the OGB team for sharing their website template for the TGB website. This research was enabled in part by compute resources, software and technical help provided by Mila. This research was supported by the Canadian Institute for Advanced Research (CIFAR AI chair program), Natural Sciences and Engineering Research Council of Canada (NSERC) Postgraduate Scholarship Doctoral (PGS D) Award and Fonds de recherche du Québec - Nature et Technologies (FRQNT) Doctoral Award.

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

# A Dataset Checklist

1. Submission introducing new datasets must include the following in the supplementary materials

   (a) Dataset documentation and intended uses. [Yes] We include the datasheets for datasets of TGB 2.0 in Appendix I.

   (b) URL to website/platform where the dataset/benchmark can be viewed and downloaded by the reviewers. [Yes] The website link and documentation link is included in Appendix C.

   (c) URL to Croissant metadata record documenting the dataset/benchmark available for viewing and downloading by the reviewers. [Yes] The croissant metadata record link is `https://object-arbutus.cloud.computecanada.ca/tgb/tgb2_croissant.json`.

   (d) Author statement that they bear all responsibility in case of violation of rights, etc., and confirmation of the data license. [Yes] Yes, we bear all responsibility and also state this in Appendix D.

   (e) Hosting, licensing, and maintenance plan. [Yes] Yes, we discuss the hosting and licensing plan in Appendix C.

2. To ensure accessibility, the supplementary materials for datasets must include the following:

   (a) Links to access the dataset and its metadata. [Yes] Yes, all links are provided in Appendix D and C.

   (b) The dataset itself should ideally use an open and widely used data format. Provide a detailed explanation on how the dataset can be read. For simulation environments, use existing frameworks or explain how they can be used. [Yes] The dataset is automatically downloaded and processed by the TGB 2.0 code and presented in ML ready format.

   (c) Long-term preservation: It must be clear that the dataset will be available for a long time, either by uploading to a data repository or by explaining how the authors themselves will ensure this. [Yes] TGB 2.0 datasets are maintained via Digital Research Alliance of Canada (funded by the Government of Canada).

   (d) Explicit license: Authors must choose a license, ideally a CC license for datasets, or an open source license for code (e.g. RL environments). [Yes] Yes, all dataset licenses are provided in Appendix D. The TGB 2.0 code is provided under the MIT license.

   (e) Add structured metadata to a dataset's meta-data page using Web standards (like schema.org and DCAT): This allows it to be discovered and organized by anyone. If you use an existing data repository, this is often done automatically. [Yes] We provide the croissant metadata record, the link is `https://object-arbutus.cloud.computecanada.ca/tgb/tgb2_croissant.json`.

   (f) Highly recommended: a persistent dereferenceable identifier (e.g. a DOI minted by a data repository or a prefix on identifiers.org) for datasets, or a code repository (e.g. GitHub, GitLab,...) for code. If this is not possible or useful, please explain why. [Yes] The DOI for the project is `https://zenodo.org/doi/10.5281/zenodo.11480521`.

# B Broader Impact

**Impact on Temporal Graph Learning.** Recently, the availability of large graph benchmarks accelerates research in the field [27, 26, 12]. By providing a standardized benchmarking framework, TGB 2.0 will accelerate the development and evaluation of new models for temporal knowledge graphs and temporal heterogeneous graphs. Researchers can build on a common foundation, leading to more rapid and robust advancements in this field. In addition, the introduction of a unified evaluation framework addresses reproducibility issues, which are critical to scientific progress. The comprehensive evaluation facilitated by TGB 2.0 ensures that new methods are rigorously tested against state-of-the-art baselines, leading to more robust and well-validated models. This contributes to higher standards in research and more reliable outcomes. Overall, this work has the potential to significantly impact both the academic research community and practical applications, driving forward the understanding and utilization of multi-relational temporal graphs in various fields.

**Potential Negative Impact.** The TGB 2.0 datasets may limit the utilization and mining of other TG datasets. If the datasets are not representative of the broader set of real-world data, this could lead to biased or unfair outcomes when models are applied in practice. Similarly, the community might become overly dependent on the TGB 2.0 framework, potentially hindering the exploration of alternative benchmarking methodologies or the development of diverse evaluation protocols that might be more suitable for specific contexts or emerging sub-fields. In addition, when the focus is mainly on quantitative performance metrics, it could overshadow the importance of qualitative assessments and other critical factors, such as interpretability, fairness, and ethical considerations in the development and implementation of models. To avoid this issue, we plan to update TGB 2.0 regularly with community feedback as well as adding additional datasets and tasks.

## C   Dataset Documentation and Intended Use

All datasets presented by TGB 2.0 are intended for academic use and their corresponding licenses are listed in Appendix D. We also anonymized the datasets, to remove any personally identifiable information where appropriate. For ease of access, we provide the following links to the TGB 2.0 benchmark suits and datasets.

- The code is available publicly on TGB2 Github: `https://github.com/JuliaGast/TGB2`. The code will also be merged into TGB Github.
- Dataset and project documentations can be found at: `https://tgb.complexdatalab.com/`.
- Tutorials and API references can be found at: `https://docs.tgb.complexdatalab.com/`.
- Hugging face link for main dataset files is `https://huggingface.co/datasets/andrewsleader/TGB/tree/main`.
- ML croissant metadata file link is `https://object-arbutus.cloud.computecanada.ca/tgb/tgb2_croissant.json`.

**Maintenance Plan.** We plan to continue to improve and develop TGB 2.0 based on community feedback to provide a reproducible, open and robust benchmark for temporal multi-relational graphs. We will maintain and improve the TGB 2.0, TGB and TGB-Baselines github repository, while the TGB 2.0 datasets are maintained via Digital Research Alliance of Canada (funded by the Government of Canada).

## D   Dataset Licenses and Download Links

In this section, we present dataset licenses and the download link (embedded in dataset name). The datasets are maintained via Digital Research Alliance of Canada funded by the Government of Canada. As authors, we confirm the data licenses as indicated below and that we bear all responsibility in case of violation of rights. We also included the metadata for datasets in the ML croissant format [2]. The ML croissant metadata link is `https://object-arbutus.cloud.computecanada.ca/tgb/tgb2_croissant.json`.

- `tkgl-smallpedia`: Wikidata License. See license information from Wikidata License Page. Property and lexeme namespaces is made available under the Creative Commons CC0 License. Text in other namespaces is made available under the Creative Commons Attribution-ShareAlike License. Here is the data source link.
- `tkgl-polecat`: CC0 1.0 DEED license. Here is the data source link.
- `tkgl-icews`: Custom Dataset License. The detailed license information can be found here. Restrictions on use: these materials are subject to copyright protection and may only be used and copied for research and educational purposes. The materials may not be used or copied for any commercial purposes. Here is the data source link.
- `tkgl-wikidata`: Wikidata License. See license information from Wikidata License Page. Property and lexeme namespaces is made available under the Creative Commons CC0 License. Text in other namespaces is made available under the Creative Commons Attribution-ShareAlike License. Here is the data source link.

- `thgl-software`: CC-BY-4.0 license. This dataset is curated from GH Arxiv code which has the MIT License. Content based on www.gharchive.org is released under the CC-BY-4.0 license. To avoid any personal identifiable information, we anonymized all nodes to integers. The raw data can be found here.

- `thgl-forum`: CC BY-NC 2.0 DEED license. The raw data source is here [54].

- `thgl-myket`: CC BY-NC 4.0 DEED license. A smaller subset of this dataset is available on Github.

- `thgl-github`: CC-BY-4.0 license.This dataset is curated from GH Arxiv code which has the MIT License. Content based on www.gharchive.org is released under the CC-BY-4.0 license. To avoid any personal identifiable information, we anonymized all nodes to integers. The raw data can be found here.

## E   Data Processing Details

Here we discuss the data collection and cleaning process for TGB 2.0 datasets in detail. For reproducibility, we also provide the corresponding mining script on Github.

**TKG Dataset Format.** For all TKG datasets, we include the temporal links in `edgelist.csv`. The validation and test negative samples are included in `val_ns.pkl` and `test_ns.pkl` files respectively. Additionally, for `tkgl-smallpedia` and `tkgl-wikidata`, we also include the static links related to the wiki entities in `static_edgelist.csv`.

**THG Dataset Format.** For all THG datasets, we include the temporal links in `edgelist.csv`. The validation and test negative samples are included in `val_ns.pkl` and `test_ns.pkl` files respectively. We also include `nodemapping.csv` and `edgemapping.csv` to provide the name of each node relation and edge relation respectively. Lastly, `nodetype.csv` provides the description for node type information.

`tkgl-wikidata` (scripts). This TKG dataset is extracted from the Wikidata KG [68] from 2024 February 20th. We extract the relations from the first 32 million Wikidata entities (by the entity ID). We then retain the relations with the temporal qualifier with relation IDs: `P585`, `P580`, `P582`, `P577`, `P574`. We also retain the static relations with these wiki entities, filtering out less informative relations including `P31` and `P279`. We then keep only links with both start and end date as well as point in time relations. We retain links starting from 0 BC. We also remove any links that show up as both static and temporal link.

`tkgl-smallpedia` (scripts). This is a subset of the `tkgl-wikidata` where links from the first 1 million entities are retained and then filtered by their year. Only links from 1900 to 2024 are kept. We also extract the static relations associated with these first 1 million entities. Further, we remove any links that show up as both static and temporal link.

`tkgl-polecat` (scripts). We extract the raw files from the POLECAT dataset source [62], we organize the monthly edgelist chronologically and then merge them into a single file. Then, we remove any links with missing source or destination.

`tkgl-icews` (scripts). We extract the raw files from ICEWS Coded Event Data source [7, 65]. We first combine the monthly files into a single one, removing any links missing a source or destination.

`thgl-github` (scripts). The raw data is extracted from GH Arxiv, containing the data from March 2024. We then extract 14 relations based on the common activities on Github. We also removed the `issue comment` and `pr review comment` node types as they are often one time nodes which rarely repeat, and kept nodes which have at least two edges in the dataset.

`thgl-software` (scripts). The raw data is extracted from GH Arxiv, containing the data from January 2024. We kept nodes with at least 10 interactions while removing the `issue comment` and `pr review comment` node types.

`thgl-forum` (scripts). The raw data is downloaded from here [54]. We first merge edge files and attribute files by edge ID. Next, we filter out nodes with less than 100 links from the dataset. The node types are user nodes and subreddit nodes. The edge types are user replying to user and user posting on subreddit.

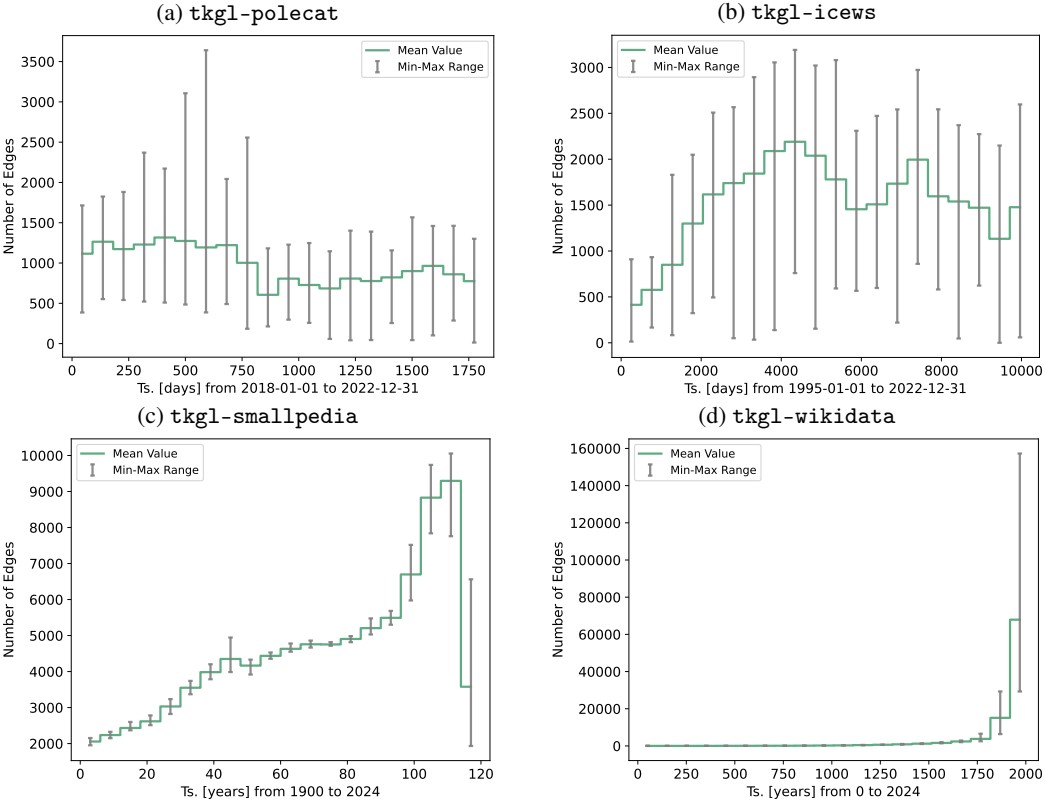

Figure 5: Dataset Edges over time for TKG.

`thgl-myket` (scripts). This is an original dataset provided by the Myket Corporation. The data indicates if an edge is a software upgrade or a first time install which forms the two edge types in this data. The node types are users and apps.

## F  Dataset Statistics

Figure 5 shows how the number of edges change over time for TKG datasets. Figures 6 shows how the number of edges change over time for THG datasets. While most datasets exhibit fluctuations in the number of edges around a constant level, `tkgl-wikidata` stands out with a significant upward trend in the number of edges over the years, indicating a surge in events, particularly in recent years. In addition, noteworthy deviations in timesteps are apparent. TKG datasets display anomalous timesteps characterized by minimal edge numbers, particularly evident during the Covid pandemic for `tkgl-icews`. Conversely, for the THG datasets the occurrence of zero-edge timesteps is not indicative of outliers; rather, it reflects the continuous nature of the data, where not every second entails an event occurrence. THG datasets exhibit instances of exceptionally high edge counts per timestep, such as in the case of `thgl-forum` with up to 120 edges per timestamp.

Figure 7 shows the top ten most frequent edge types in TKG datasets. Figure 8 shows the top ten most frequent edge types in THG datasets. Note that TKG datasets in general have more edge types than THG datasets. Most common THG relations usually share similar portion of edges in the dataset while TKG relations shares different portion of edges.

## G  Experimental Details

In the following, we provide additional experimental details such as the computing resources, resource consumption, hyperparameters, and runtime statistics.

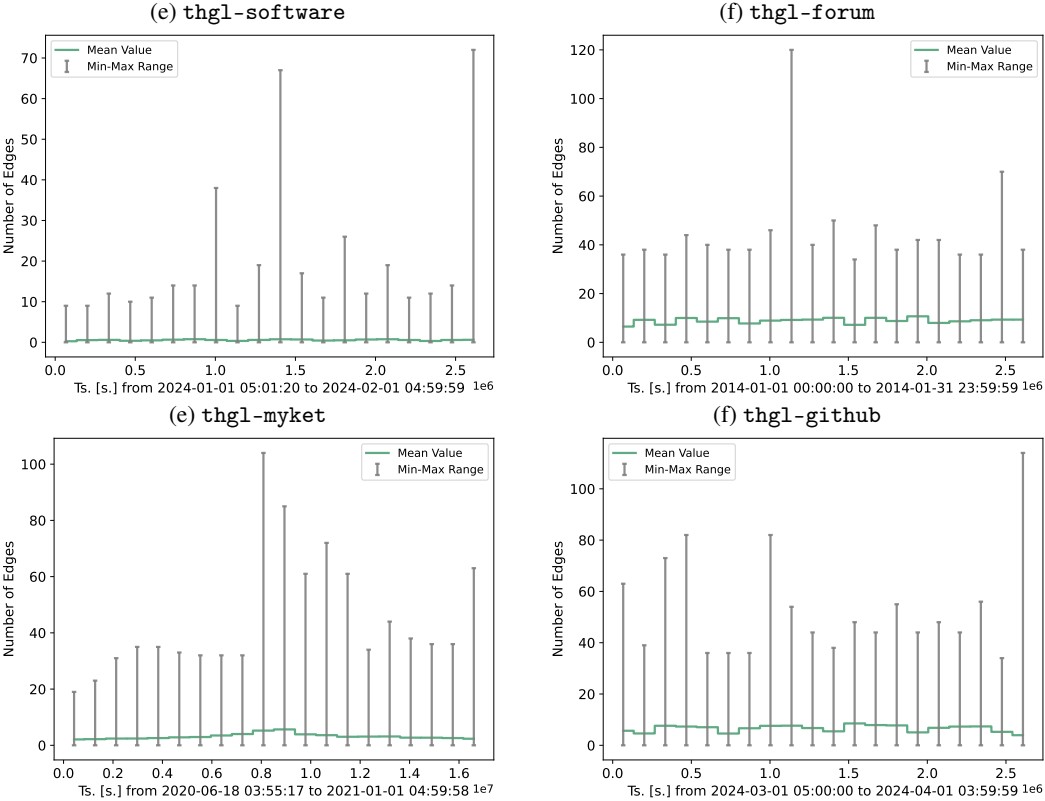

Figure 6: Dataset Edges over time for THG.

## G.1 Computing Resources

We ran all experiments on either Narval or Béluga cluster of Digital Research Alliance of Canada or the Mila, Québec AI Institute cluster. For the experiments on the Narval cluster, we ran each experiment on a Nvidia A100 (40G memory) GPU with 4 CPU nodes (from either of the AMD Rome 7532 @ 2.40 GHz 256M cache L3, AMD Rome 7502 @ 2.50 GHz 128M cache L3, or AMD Milan 7413 @ 2.65 GHz 128M cache L3 available type) each with 100GB memory. For experiments on the Béluga cluster, we ran each experiments on a NVidia V100SXM2 (16G memory) GPU wiht 4 CPU nodes (from Intel Gold 6148 Skylake @ 2.4 GHz) each with 100GB memory. For the experiments on the Mila cluster, we ran each experiment on an RTX8000 (40G memory) GPU or an V100 (32G memory) GPU with 4 CPU nodes (from either of the AMD Rome 7532 @ 2.40 GHz 256M cache L3, AMD Rome 7502 @ 2.50 GHz 128M cache L3, or AMD Milan 7413 @ 2.65 GHz 128M cache L3 available type). The upper limit of RAM was set to 1056GB.

A seven-day time limit was considered for each experiment. For all non deterministic methods, i.e. all methods besides Edgebank and the Recurrency Baseline, we repeated each experiments five times and reported the average and standard deviation of different runs. It is noteworthy that except for the reported baseline results, the other models, all evaluated by their original source code, throw an out of memory error or do not finish in the given time limit for the medium and large datasets on all available resources including Narval, Béluga, and Mila clusters.

## G.2 GPU Usage Comparison

In Table 4 and 5, we report the average GPU usage of TKG and THG methods on the dataset across 5 trials. Note that the Recurrency Baseline, EdgeBank, and TLogic only require CPU thus no GPU usage is reported. For TKG, some methods such as CEN on `tkgl-polecat` have higher GPU usage when compared to others. For THG, scalability is a significant issue, as most methods involve high GPU usage and often result in out-of-memory errors, especially with larger datasets. Although STHN

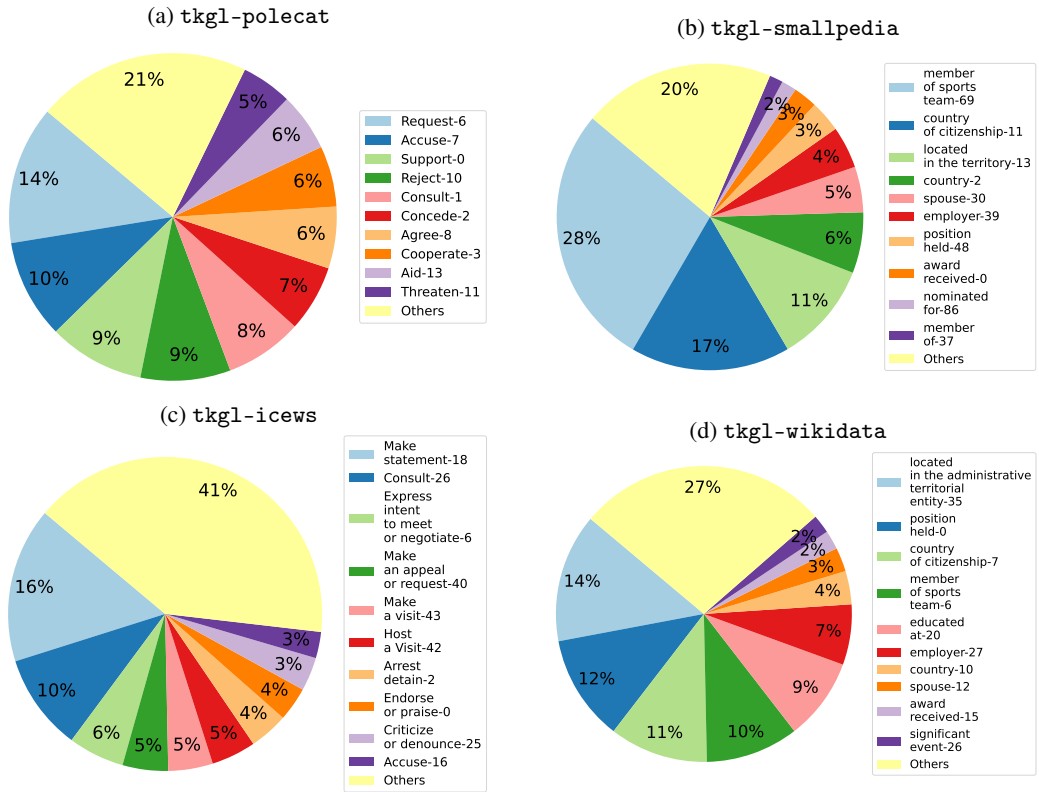

Figure 7: Edge type ratios in TGB 2.0 TKGs. We include the 10 most frequent edge types.

Table 4: GPU memory usage in **GB** for the *Temporal Knowledge Graph Link Prediction* task for the methods that run on GPU. We report the average across 5 runs.

| Method | tkgl-smallpedia | tkgl-polecat | tkgl-icews | tkgl-wikidata |
|---|---|---|---|---|
| RE-GCN [41] | 20.9 | 21.2 | 24.3 | OOM |
| CEN [39] | 28.8 | 41.0 | 31.6 | OOM |

maintains manageable GPU usage, it requires substantial RAM to compute the subgraphs, making it impractical for use in all environments.

## G.3   Runtime Comparison

In Table 6 and Table 7 we report the inference times as well as the total time for training, validation and testing for each method for TKG and THG experiments. For the non-deterministic methods, we report the average across 5 runs. The tables illustrate that both, inference times, as well as total times vary significantly across methods.

## G.3.1   Hyperparameters

If not stated otherwise, for each method we use the hyperparameter setting as reported in the original papers, please see Table 8. Whereas further hyperparameter tuning could further improve performance of each method, it was out of scope for this work. We only change the hyperparameter values if the methods would not finish with the given time or memory limit. In this case, we follow recommendations from [16] (to decrease rule length and window size for TLogic), from the authors of [15] (to decrease the window length for the Recurrency Baseline), and from the authors of [41] (to decrease the history length for RE-GCN and CEN).

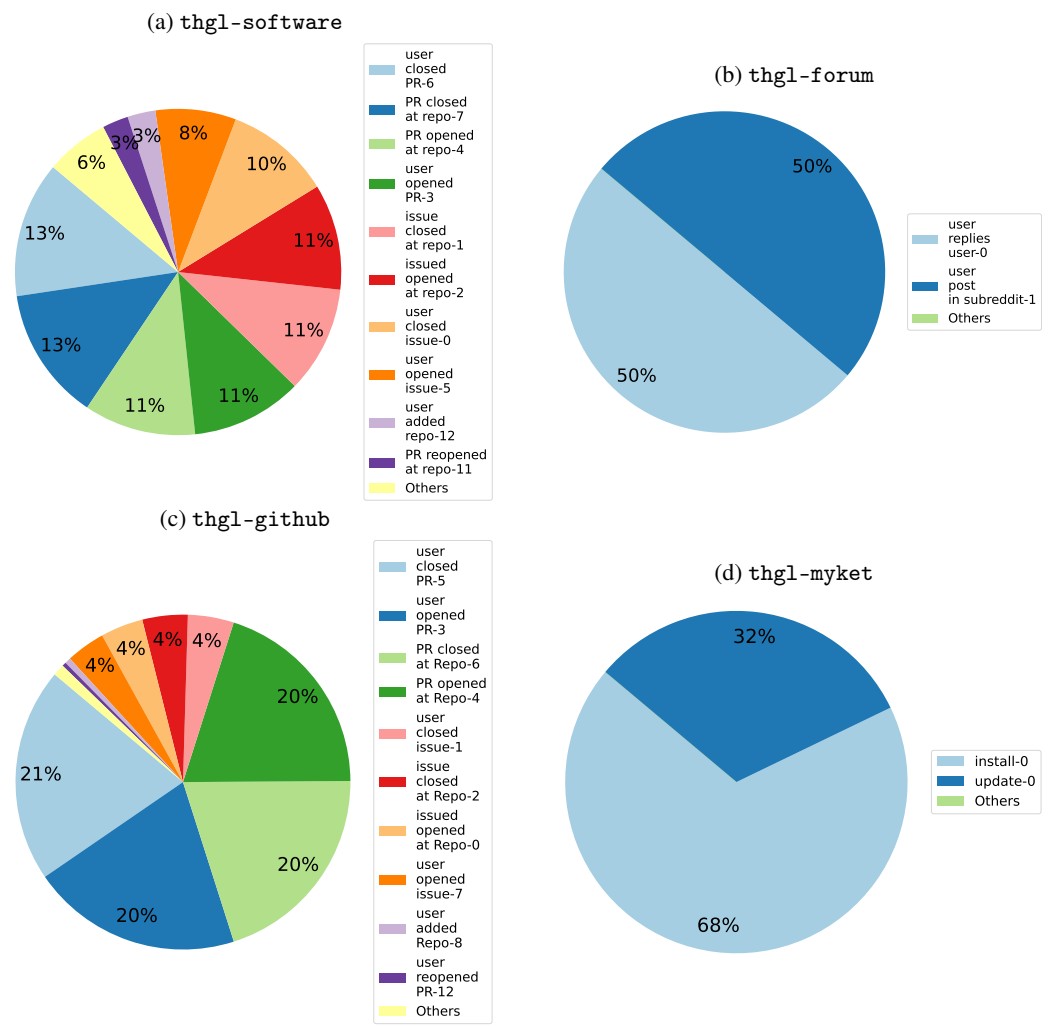

Figure 8: Edge type ration in TGB 2.0 THGs.

Table 5: GPU memory usage in **GB** for *Temporal Heterogeneous Graph Link Prediction* task. We report the average across 5 runs.

| Method | `thgl-software` | `thgl-forum` | `thgl-myket` | `thgl-github` |
|---|---|---|---|---|
| TGN [61] | 7 | 8 | - | - |
| TGN$_{\text{edge-type}}$ | 10 | 12 | - | - |
| STHN [38] | 15 | - | - | - |

## G.4 Experimental Observations

Several methods encountered memory limitations or did not complete within the designated time constraints. Thus, as described in Section 5, their performance is not reported. In the following, we provide additional details on the problems of individual methods:

- RE-GCN and CEN run out of GPU memory for `tkgl-wikidata`, even if severely limiting embedding dimension and history length.

- Recurrency Baseline does not finish in the designated time constraint for the large THG datasets `thgl-myket` and `thgl-github` and the large TKG dataset `tkgl-wikidata`.

Table 6: Inference time as well as total train and validation times for *Temporal Knowledge Graph Link Prediction* task in **seconds**. For non-deterministic methods, we report the average across 5 different runs.

| Method | tkgl-smallpedia | | tkgl-polecat | | tkgl-icews | | tkgl-wikidata | |
|---|---|---|---|---|---|---|---|---|
| | Test | Total | Test | Total | Test | Total | Test | Total |
| EdgeBank$_{tw}$ [57] | 2,935 | 5,810 | 46,629 | 94,475 | 311,278 | 600,929 | 5,445 | 8,875 |
| EdgeBank$_\infty$ [57] | 4,417 | 8,259 | 31,713 | 64,157 | 203,268 | 412,774 | 4,814 | 7,923 |
| RecurrencyBaseline$_{train}$ [15] | 310 | 9,895 | 3,392 | 80,378 | 3,928 | 148,710 | - | - |
| RecurrencyBaseline$_{default}$ [15] | 316 | 659 | 4,500 | 8,343 | 11,756 | 30,110 | - | - |
| RE-GCN [41] | 165 | 3,895 | 1,766 | 45,877 | 6,848 | 114,370 | - | - |
| CEN [39] | 331 | 14,493 | 2,726 | 77,953 | 8,999 | 202,477 | - | - |
| TLogic [47] | 331 | 803 | 75,654 | 138,636 | 60,413 | 128,391 | - | - |

Table 7: Inference time as well as total train and validation time for *Temporal Heterogeneous Graph Link Prediction* task in **seconds**. For the non-deterministic methods, we report the average across 5 different runs.

| Method | thgl-software | | thgl-forum | | thgl-myket | | thgl-github | |
|---|---|---|---|---|---|---|---|---|
| | Test | Total | Test | Total | Test | Total | Test | Total |
| EdgeBank$_{tw}$ [57] | 102 | 203 | 1,158 | 2,329 | 4,820 | 9,603 | 295 | 301 |
| EdgeBank$_\infty$ [57] | 107 | 212 | 1,148 | 2,303 | 4,956 | 10,017 | 282 | 296 |
| RecurrencyBaseline$_{default}$ [15] | 62,259 | 114,124 | 32,539 | 65,114 | - | - | - | - |
| TGN [61] | 686 | 66,290 | 7,654 | 8,8659 | - | - | - | - |
| TGN$_{edge-type}$ | 567 | 39,427 | 8,241 | 111,494 | - | - | - | - |
| STHN [38] | 52,101 | 102,943 | - | - | - | - | - | - |

- TLogic does not finish in the designated time constraint for `tkgl-wikidata`. Further, we reduced the rule length to 1 to fit in the time constraint and memory limitations for the introduced datasets.

- STHN model has very high memory consumption, requires 185 GB of RAM on the small `thgl-software` dataset (mostly due to subgraph computations). On the rest of THG datasets, it rans out of memory.

- TGN and TGN$_{edge-type}$ run out of GPU memory for both `thgl-myket` and `thgl-github`, even if limiting embedding dimension to time_dim $=$ mem_dim $=$ emb_dim $= 16$ and edgeType_dim $= 16$.

Table 8: Hyperparameter choices for each method for All datasets. Values that are different from the original papers are **bolded**. In case we modify the values for different datasets, we report so in the column *Dataset-specific*.

| Method | Hyperparameter Values | |
|---|---|---|
| | All Datasets | Dataset-specific |
| TLogic | **rule_lengths = 1**, window $= 0$, top_k $= 20$ | `tkgl-icews`: window $= 500$ |
| RE-GCN | n_hidden $= 200$, n_layers $= 2$, dropout $= 0.2$, lr $= 0.001$, n_bases $= 100$, train_history_len $= 3$, test_history_len $= 3$ | |
| CEN | n_hidden $= 200$, n_layers $= 2$, dropout $= 0.2$, lr $= 0.001$, n_bases $= 100$, n_layers $= 2$, **train_history_len** $= 3$ test_history_len $= 3$, **start_history_len** $= 2$, dilate_len $= 1$ | |
| RecB$_{default}$ | $\lambda = 0.1$, $\alpha = 0.99$, window $= 0$ | `tkgl-icews`: window $= 500$ |
| RecB$_{train}$ | $\lambda = 0.1$, $\alpha = 0.99$, window $= 0$ | `tkgl-icews` : window $= 100$ |
| TGN | lr $= 1e-04$, mem_dim $= 100$, time_dim $= 100$, emb_dim $= 100$, num_neighbors $= 10$ | |
| TGN$_{edge-type}$ | lr $= 1e-04$, mem_dim $= 100$, time_dim $= 100$, emb_dim $= 100$, num_neighbors $= 10$, edge_emb_dim $= 16$ | |
| STHN | lr $= 5e-04$, max_edges $= 50$, window_size $= 5$, dropout $= 0.1$, time_dims $= 100$, hidden_dims $= 100$ | |

Table 9: MRR and Runtime for Edgebank and the Recurrency Baseline (RecB) on the `tkgl-smallpedia` dataset for three different strategies for Negative Sample Generation.

| Strategy | Method | MRR | | Runtime [s.] | |
|---|---|---|---|---|---|
| | | valid | test | test | total |
| 1-vs-1000 (random) | RecB$_{default}$ [15] | 0.755 | 0.734 | 278 | 692 |
| | EdgeBank$_{tw}$ [57] | 0.706 | 0.576 | 72 | 141 |
| 1-vs-1000 (ours) | RecB$_{default}$ [15] | 0.642 | 0.608 | 282 | 703 |
| | EdgeBank$_{tw}$ [57] | 0.612 | 0.495 | 104 | 210 |
| 1-vs-all | RecB$_{default}$ [15] | 0.542 | 0.486 | 316 | 659 |
| | EdgeBank$_{tw}$ [57] | 0.457 | 0.353 | 2935 | 5810 |

## G.5 Ablation Study on Negative Sample Generation

Here, we compare results for evaluation on the full set of nodes (*1-vs-all*) versus a limited number of negative samples $q$ (*1-vs-q*). We also compare our sampling method based on destination nodes of each edge type (*1-vs-q (ours)*) with that of random sampling (*1-vs-q (random)*). We select the `tkgl-smallpedia` dataset and report results for the Recurrency Baseline as well as Edgebank, as both methods perform competitively while being deterministic. Table 9 confirms expectations: random negative sampling yields the highest MRR values. MRR values for our destination-aware negative sampling demonstrate a closer proximity to the full sampling (*1-vs-all*) for both methodologies. Notably, employing the 1-vs-all approach yields the lowest MRR for both test and validation sets, underscoring the importance of comprehensive evaluations whenever feasible. However, particularly evident in the case of Edgebank, the adoption of negative sampling significantly reduces test time, changing from approximately 3000 seconds to 70 seconds.

## G.6 Details on Evaluation Protocol

As described in section 1, we compute the time-aware *Mean Reciprocal Rank* [16]. Specifically, for each test edge $(s_{test}, r_{test}, o_{test}, t_{test})$, we evaluate the models prediction by removing the object $o_{test}$ from the test query, $(s_{test}, r_{test}, ?, t_{test})$. The model then assigns scores to all possible entities for the object position, which are subsequently ranked in descending order. This is repeated for subject prediction. The MRR is calculated as the mean of the reciprocal of these ranks across all test queries. In applying the *time-aware filter setting*, we filter out quadruples that have the same timestamp as the test query. For example, for a test query (France, wins, Basketball Match, 2025-05-19) a model's prediction (France, wins, Soccer Match, 2025-05-19) would be excluded if (France, wins, Soccer Match, 2025-05-19) is present in the test set. However, it would not be filtered out if (France, wins, Soccer Match, 2025-03-09) is present in the test set.

We evaluate all methods in *single-step prediction*. This means that the model always forecasts the next timestep, and then ground truth facts are fed before predicting the subsequent timestep [16]. If there are multiple triples in the candidate set with the same score from the model, our protocol assigns the average rank, i.e. the average between the worst (pessimistic) and best (optimistic) score, to the ground truth triple.

The scripts for generating the negative samples can be found here under the folder for each individual dataset [4]. We select the following negative sampling strategies for each dataset:

- `tkgl-smallpedia`, `tkgl-polecat`, `tkgl-icews`: *1-vs-all*

- `tkgl-wikidata`, `thgl-software`: *1-vs-q* with $q = 1000$

- `thgl-github`, `thgl-myket`: *1-vs-q* with $q = 20$

- `thgl-forum`: *1-vs-q* with $q = 100$

---

[4]the scripts all have suffix of `ns_gen.py`

Table 10: Comparison of test results when using Hits at 10 (H@10) vs. MRR metric for *Temporal Knowledge Graph Link Prediction* task.

| Method
Test Score | tkgl-smallpedia
MRR | 
H@10 | tkgl-polecat
MRR | 
H@10 | tkgl-icews
MRR | 
H@10 | tkgl-wikidata
MRR | 
H@10 |
|---|---|---|---|---|---|---|---|---|
| EdgeBank$_{tw}$ [57] | 0.353 | 0.566 | 0.056 | 0.119 | 0.020 | 0.058 | 0.535 | 0.596 |
| EdgeBank$_\infty$ [57] | 0.333 | 0.562 | 0.045 | 0.094 | 0.009 | 0.014 | 0.535 | 0.596 |
| RecurrencyBaseline$_{train}$ [15] | 0.605 | **0.716** | 0.198 | 0.317 | **0.211** | 0.324 | - | - |
| RecurrencyBaseline$_{default}$ [15] | 0.486 | 0.651 | 0.167 | 0.264 | 0.206 | 0.324 | - | - |
| RE-GCN [41] | 0.594 | 0.687 | 0.175 | 0.292 | 0.182 | 0.331 | - | - |
| CEN [39] | **0.612** | 0.705 | 0.184 | 0.323 | 0.187 | **0.334** | - | - |
| TLogic [47] | 0.595 | 0.707 | **0.228** | **0.378** | 0.186 | 0.301 | - | - |

Table 11: Number of Edges and timestamps for train, validation and test set for each dataset in TGB 2.0.

| Dataset | Temporal Knowledge Graphs (tkgl-) | | | | Temporal Heterogeneous Graphs (thgl-) | | | |
|---|---|---|---|---|---|---|---|---|
| | smallpedia | polecat | icews | wikidata | software | forum | github | myket |
| # Train Quadruples | 387,757 | 1,246,556 | 10,861,600 | 6,982,503 | 1,042,866 | 16,630,396 | 12,249,711 | 37,542,951 |
| # Valid Quadruples | 81,033 | 266,736 | 2,326,157 | 1,434,950 | 223,469 | 3,563,658 | 2,624,934 | 8,044,922 |
| # Test Quadruples | 81,586 | 266,318 | 2,325,689 | 1,438,750 | 223,471 | 3,563,653 | 2,624,932 | 804,4915 |
| # All Quadruples | 550,376 | 1,779,610 | 15,513,446 | 9,856,203 | 1,489,806 | 23,757,707 | 17,499,577 | 53,632,788 |
| # Train Timesteps | 98 | 1,193 | 7,187 | 1,999 | 485,863 | 1,805,376 | 1,703,696 | 9,935,183 |
| # Valid Timesteps | 10 | 329 | 1,341 | 12 | 99,500 | 393,000 | 382,882 | 2,274,936 |
| # Test Timesteps | 17 | 304 | 1,696 | 14 | 104,186 | 360,081 | 423,837 | 2,617,971 |
| # All Timesteps | 125 | 1,826 | 10,224 | 2,025 | 689,549 | 2,558,457 | 2,510,415 | 14,828,090 |

## G.7 Results on Additional Metrics

In addition to the MRR, in Table 10, we report the Hits at 10 (H@10), the proportion of test queries for which at least one correct item is among the top 10 ranked results. The results show that the ranking of methods in Hits@10 and MRR metric are closely, but not fully matched.

## G.8 Detailed information on Train, Validation, and Test Splits

As described in Section 4, we split all datasets chronologically into the training, validation, and test sets, respectively containing $70\%$, $15\%$, and $15\%$ of all edges. Because we ensure that edges for a timestep can only be in either train or validation or test set, and because the number of edges over time are not constant, the cuts are not strict. We provide more details on the exact splits in Table 11.

## H   More Details on Methods

In the following we will describe the methods that we selected for our experiments.

### H.1   Temporal Knowledge Graph Forecasting

For our experiments we select methods from a variety of methods from the previous literature. We base our selection on a) code availability, b) comparatively high performance in previous studies on smaller datasets (following results as reported in [16] and [15], i.e. we exclude methods that are reported to have lower MRRs on all previous datasets as compared to the Recurrency Baseline), and c) we exclude methods that have reported to have long runtimes or high GPU memory consumption on the existing smaller datasets (e.g. [22] for the GDELT dataset [16]). This results in the following TKG baselines:

- *RE-GCN [41]* learns from the sequence of Knowledge Graph snapshots recurrently by combining a convolutional graph Neural Network with a sequential Neural Network model. It also incorporates a static graph constraint to include additional information like entity types.
- *CEN [39]* integrates a GCN capable of handling evolutional patterns of different lengths through a learning strategy that progresses from short to long patterns. This model can

adapt to changes in evolutional patterns over time in an online setting, being updated with historical facts during testing.

- *TLogic [47]* is a symbolic framework that learns temporal logic rules via temporal random walks, traversing edges backward in time through the graph. It applies these rules to events preceding the query, considering both the confidence of the rules and the time differences for scoring answer candidates.

- Recurrency Baseline [15] is a baseline method that predicts recurring facts by combining scores based on strict recurrency, considering the recency and frequency of these facts, and scores based on relaxed recurrency, which accounts for the recurrence of parts of the query. Two versions of this baseline are tested: $RecB_{default}$, which uses default parameter values, and $RecB_{train}$, which selects parameter values based on a grid search considering performance on the validation set.

## H.2  Temporal Heterogeneous Graph Forecasting

- *TGN [61]* represents a comprehensive framework designed for learning on dynamic graphs in continuous time. Its components include a memory module, message function, message aggregator, memory updater, and embedding module. During testing, TGN updates the memories of nodes with edges that have been newly observed. Additionally, to incorporate edge types into TGN, we devised a variant of TGN capable of utilizing edge type information. This was achieved by generating embeddings from the edge types, which were then concatenated with the original messages within the TGN model.

- *STHN [38]* designed for continuous-time link prediction on Temporal heterogeneous networks that efficiently manages dynamic interactions. The architecture consists of a *Heterogeneous Link Encoder* with type and time encoding components, which embed historical interactions to produce a temporal link representation. The process continues with *Semantic Patches Fusion*, where sequential representations are divided into different patches treated as token inputs for the Encoder, and average mean pooling compresses these into a single vector. Finally, the framework combines the representations of nodes $u$ and $v$, utilizing a fully connected layer and *CrossEntropy* loss for link prediction, effectively capturing complex temporal information and long-term dependencies.

# I  Datasheets for Datasets

This section answers questions about this work based on Datasheets for Datasets [17].

### I.0.1  Motivation

- **For what purpose was the dataset created?** Was there a specific task in mind? Was there a specific gap that needed to be filled? Please provide a description. TGB 2.0 is curated for realistic, reproducible and robust evaluation for temporal multi-relational graphs. Specifically there are four TKG datasets and four THG datasets, all designed for the dynamic link property prediction task.

- **Who created the dataset (e.g., which team, research group) and on behalf of which entity (e.g., company, institution, organization)?** `thgl-software` and `thgl-github` datasets are based on Github data collected by GH Arxiv. `thgl-forum` dataset is derived from user and subreddit interactions on Reddit. `thgl-myket` dataset was generated by the data team of the Myket Android application market. `tkgl-smallpedia` and `tkgl-wikidata` datasets are constructed from the Wikidata Knowledge Graph. `tkgl-polecat` is based on the POLitical Event Classification, Attributes, and Types (POLE-CAT) dataset. `tkgl-icews` is extracted from the ICEWS Coded Event Data. Detailed Dataset information is found in Section 4.

- **Who funded the creation of the dataset?** If there is an associated grant, please provide the name of the grantor and the grant name and number. Funding information is provided in Acknowledgement Section.

### I.0.2 Composition

- **What do the instances that comprise the dataset represent (e.g., documents, photos, people, countries)?** Are there multiple types of instances (e.g., movies, users, and ratings; people and interactions between them; nodes and edges)? Please provide a description.

  The datasets primarily consist of nodes and edges in graph structures, representing various entities and their interactions:

  - `thgl-software` **and** `thgl-github`**:** Nodes represent entities like users, pull requests, issues, and repositories. Edges indicate interactions among these entities.
  - `thgl-forum`**:** Comprises user and subreddit nodes with edges for user replies and posts.
  - `thgl-myket`**:** Features nodes as users and Android applications, with edges detailing install and update interactions. These datasets facilitate tasks like predicting future interactions or activities, utilizing a graph model to depict relationships in various domains such as software development, online communities, and socio-political contexts.
  - `tkgl-smallpedia` **and** `tkgl-wikidata`**:** Includes Wikidata entities as nodes with edges as temporal and static relations.
  - `tkgl-polecat` **and** `tkgl-icews`**:** Focus on socio-political actors as nodes with edges representing coded interactions.

- **How many instances are there in total (of each type, if appropriate)?** The detailed dataset statistics can be found in Section 4, Table 1.

- **Does the dataset contain all possible instances or is it a sample (not necessarily random) of instances from a larger set?** If the dataset is a sample, then what is the larger set? Is the sample representative of the larger set (e.g., geographic coverage)? If so, please describe how this representativeness was validated/verified. If it is not representative of the larger set, please describe why not (e.g., to cover a more diverse range of instances, because instances were withheld or unavailable).

  The datasets are curated from the raw source. In some cases, some data filtering is done to remove low degree nodes. More details on dataset curation is found in Section 4. For `thgl-myket`, the data provider first focused on users interacting with the platform within a two-week period and randomly sampled 1/3 of the users. The install and update interactions for these users were then tracked for three months before and after the two-week period.

- **What data does each instance consist of?** "Raw" data (e.g., unprocessed text or images)or features? In either case, please provide a description.

  The data contains the multi-relational temporal graph structure in the form of csv files as well as pre-generated negative samples for reproducible evaluation.

- **Is there a label or target associated with each instance?** If so, please provide a description.

  We focus on the dynamic link property prediction (or link prediction) task thus the goal is to predict edges in the graph in the future. Therefore, no specific task labels are provided. We also provide both node and edge type information for THGs and edge type information for TKGs.

- **Is any information missing from individual instances?** If so, please provide a description, explaining why this information is missing (e.g., because it was unavailable). This does not include intentionally removed information, but might include, e.g., redacted text.

  No, we provide information required for ML on temporal graphs.

- **Are relationships between individual instances made explicit (e.g., users' movie ratings, social network links)?** If so, please describe how these relationships are made explicit.

  The dataset themselves are classfied into TKG or THG datasets, specified by the prefix `tkgl` or `thgl`. The relations between nodes are assigned with an edge type which is provided in the csv file.

- **Are there recommended data splits (e.g., training, development/validation, testing)?** If so, please provide a description of these splits, explaining the rationale behind them.

  Yes, the recommended split uses a 70/15/15 split, and the data is split chronologically. Pleasee see Table 11 for details on the dataset splits.

- **Are there any errors, sources of noise, or redundancies in the dataset?** If so, please provide a description.

  No. However, datasets such as `tkgl-smallpedia` and `tkgl-wikidata` are extracted from Wikipedia where the knowledge is crowd-sourced, and thus may contain errors.

- **Is the dataset self-contained, or does it link to or otherwise rely on external resources (e.g., websites, tweets, other datasets)?** If it links to or relies on external resources, a) are there guarantees that they will exist, and remain constant, over time; b) are there official archival versions of the complete dataset (i.e., including the external resources as they existed at the time the dataset was created); c) are there any restrictions (e.g., licenses, fees) associated with any of the external resources that might apply to a dataset consumer? Please provide descriptions of all external resources and any restrictions associated with them, as well as links or other access points, as appropriate.

  The dataset is self-contained.

- **Does the dataset contain data that might be considered confidential (e.g., data that is protected by legal privilege or by doctor–patient confidentiality, data that includes the content of individuals' nonpublic communications)?** If so, please provide a description.

  No, all data are gathered from public sources and we have anonymized user information where appropriate.

- **Does the dataset contain data that, if viewed directly, might be offensive, insulting, threatening, or might otherwise cause anxiety?** If so, please describe why.

  No.

- **Does the dataset identify any subpopulations (e.g., by age, gender)?** If so, please describe how these subpopulations are identified and provide a description of their respective distributions within the dataset.

  No.

- **Is it possible to identify individuals (i.e., one or more natural persons), either directly or indirectly (i.e., in combination with other data) from the dataset?** If so, please describe how.

  No, we have anonmyzied users' information where appropriate.

- **Does the dataset contain data that might be considered sensitive in any way (e.g., data that reveals race or ethnic origins, sexual orientations, religious beliefs, political opinions or union memberships, or locations; financial or health data; biometric or genetic data; forms of government identification, such as social security numbers; criminal history)?** If so, please provide a description.

  No.

### I.0.3   Collection Process

- **How was the data associated with each instance acquired?** Was the data directly observable (e.g., raw text, movie ratings), reported by subjects (e.g., survey responses), or indirectly inferred/derived from other data (e.g., part-of-speech tags, model-based guesses for age or language)? If the data was reported by subjects or indirectly inferred/derived from other data, was the data validated/verified? If so, please describe how.

  The data is extracted from online public data sources. The data described different relations between entities. The data sources are found in Appendix D and dataset details are in Section 4.

- **What mechanisms or procedures were used to collect the data (e.g., hardware apparatuses or sensors, manual human curation, software programs, software APIs)?** How were these mechanisms or procedures validated? Software APIs.

  The datasets are curated via Python scripts written by authors, these can be found on the project Github.

- **If the dataset is a sample from a larger set, what was the sampling strategy (e.g., deterministic, probabilistic with specific sampling probabilities)?**

  For `thgl-myket`, the users where selected randomly among the users that have interactions with the platform in a two-week period. For `tkgl-smallpedia`, `tkgl-wikidata`, the

dataset was filtered by Wiki page ID. For `thgl-software` and `thgl-github`, nodes with low degrees are filtered out.

- **Who was involved in the data collection process (e.g., students, crowdworkers, contractors) and how were they compensated (e.g., how much were crowdworkers paid)?**

  Datasets are obtained from public online sources. For `thgl-myket` dataset, the interaction record of users of the platform were collected, anonymized without any personal identifiers, the data collection is discussed in the applications' privacy document. No crowdworkers are involved.

- **Over what timeframe was the data collected?** Does this timeframe match the creation timeframe of the data associated with the instances (e.g., recent crawl of old news articles)? If not, please describe the timeframe in which the data associated with the instances was created.

  Dataset timeframe and details are in Section 4.

- **Were any ethical review processes conducted (e.g., by an institutional review board)?** If so, please provide a description of these review processes, including the outcomes, as well as a link or other access point to any supporting documentation.

  No.

- **Did you collect the data from the individuals in question directly, or obtain it via third parties or other sources (e.g., websites)?**

  All datasets are obtained via websites except for `thgl-myket` which were provided by the the Myket Android application market team. Links to data sources are in Appendix D.

- **Were the individuals in question notified about the data collection?** If so, please describe (or show with screenshots or other information) how notice was provided, and provide a link or other access point to, or otherwise reproduce, the exact language of the notification itself.

  All datasets are curated from existing sources except `thgl-myket`. The data collection was discussed in the applications' privacy document.

- **Did the individuals in question consent to the collection and use of their data?** If so, please describe (or show with screenshots or other information) how consent was requested and provided, and provide a link or other access point to, or otherwise reproduce, the exact language to which the individuals consented.

  We use public data sources where data is already collected. The data collection was discussed in the applications' privacy document.

- **If consent was obtained, were the consenting individuals provided with a mechanism to revoke their consent in the future or for certain uses?** If so, please provide a description, as well as a link or other access point to the mechanism (if appropriate).

  [N/A]

- **Has an analysis of the potential impact of the dataset and its use on data subjects (e.g., a data protection impact analysis) been conducted?** If so, please provide a description of this analysis, including the outcomes, as well as a link or other access point to any supporting documentation.

  No, however the datasets are for temporal graph research purposes only, they are used to benchmark existing methods and have been anonymized appropriately.

### I.0.4   Preprocessing/cleaning/labeling

- **Was any preprocessing/cleaning/labeling of the data done (e.g., discretization or bucketing, tokenization, part-of-speech tagging, SIFT feature extraction, removal of instances, processing of missing values)?** If so, please provide a description. If not, you may skip the remaining questions in this section. No.

- **Was the "raw" data saved in addition to the preprocessed/cleaned/labeled data (e.g., to support unanticipated future uses)?** If so, please provide a link or other access point to the "raw" data.

  [N/A]

- **Is the software that was used to preprocess/clean/label the data available?** If so, please provide a link or other access point.

  [N/A]

### I.0.5 Uses

- **Has the dataset been used for any tasks already?** If so, please provide a description.

  Yes, all datasets have been tested and benchmarked in this work, see Section 5.

- **Is there a repository that links to any or all papers or systems that use the dataset?** If so, please provide a link or other access point.

  Yes, all paper references are provided in this paper. All data sources are discussed in Appendix D.

- **What (other) tasks could the dataset be used for?**

  The THG datasets can be used for other tasks such as user churn prediction and more. The TKG datasets can be used to study how knowledge changes over time.

- **Is there anything about the composition of the dataset or the way it was collected and preprocessed/cleaned/labeled that might impact future uses?** For example, is there anything that a dataset consumer might need to know to avoid uses that could result in unfair treatment of individuals or groups (e.g., stereotyping, quality of service issues) or other risks or harms (e.g., legal risks, financial harms)? If so, please provide a description. Is there anything a dataset consumer could do to mitigate these risks or harms?

  No, the datasets are for benchmarking purposes only and for researchers.

- **Are there tasks for which the dataset should not be used?** If so, please provide a description.

  No and we discuss potential negative impacts in Appendix B.

### I.0.6 Distribution

- **Will the dataset be distributed to third parties outside of the entity (e.g., company, institution, organization) on behalf of which the dataset was created?** If so, please provide a description.

  The dataset is released to the public for benchmarking on TKGs and THGs.

- **How will the dataset will be distributed (e.g., tarball on website, API, GitHub)?** Does the dataset have a digital object identifier (DOI)?

  Yes, the DOI for the project is `https://zenodo.org/records/11480522` (will point to all future version as well). The dataset download links are provided in Appendix D. TGB 2.0 datasets are maintained via Digital Research Alliance of Canada (funded by the Government of Canada).

- **When will the dataset be distributed?** The dataset is already publicly available.

- **Will the dataset be distributed under a copyright or other intellectual property (IP) license, and/or under applicable terms of use (ToU)?** If so, please describe this license and/or ToU, and provide a link or other access point to, or otherwise reproduce, any relevant licensing terms or ToU, as well as any fees associated with these restrictions. The dataset licenses are listed in Appendix D.

- **Have any third parties imposed IP-based or other restrictions on the data associated with the instances?** If so, please describe these restrictions, and provide a link or other access point to, or otherwise reproduce, any relevant licensing terms, as well as any fees associated with these restrictions. All license terms are discussed in Appendix D.

- **Do any export controls or other regulatory restrictions apply to the dataset or to individual instances?** If so, please describe these restrictions, and provide a link or other access point to, or otherwise reproduce, any supporting documentation. No.

### I.0.7 Maintenance

- **Who will be supporting/hosting/maintaining the dataset?** TGB 2.0 datasets are maintained via Digital Research Alliance of Canada (funded by the Government of Canada).

- **How can the owner/curator/manager of the dataset be contacted (e.g., email address)?** The curator of the dataset (Shenyang Huang) can be contacted via email: `shenyang.huang@mail.mcgill.ca`

- **Is there an erratum?** If so, please provide a link or other access point. No

- **Will the dataset be updated (e.g., to correct labeling errors, add new instances, delete instances)?** If so, please describe how often, by whom, and how updates will be communicated to dataset consumers (e.g., mailing list, GitHub)? Yes, the datasets will be updated based on community feedback, mainly via the main TGB Github issues.

- **If the dataset relates to people, are there applicable limits on the retention of the data associated with the instances (e.g., were the individuals in question told that their data would be retained for a fixed period of time and then deleted)?** If so, please describe these limits and explain how they will be enforced. No.

- **Will older versions of the dataset continue to be supported/hosted/maintained?** If so, please describe how. If not, please describe how its obsolescence will be communicated to dataset consumers. Any new dataset version will be annouced on Github and the TGB website.

- **If others want to extend/augment/build on/contribute to the dataset, is there a mechanism for them to do so?** If so, please provide a description. Will these contributions be validated/verified? If so, please describe how. If not, why not? Is there a process for communicating/distributing these contributions to dataset consumers? If so, please provide a description.

  Yes, first they can reach out by email to `shenyang.huang@mail.mcgill.ca` or raise a Github issue.

