# OpenReview forum: "TGB 2.0: A Benchmark for Learning on Temporal Knowledge Graphs and Heterogeneous Graphs"
_NeurIPS.cc/2024/Datasets_and_Benchmarks_Track — NeurIPS 2024 Track Datasets and Benchmarks Poster_

### Official Review · Reviewer_NhyZ · 2024-07-18
**A Benchmark for Learning on Temporal Knowledge Graphs and Heterogeneous Graphs**

**Rating:** 7
**Confidence:** 3

**Review:**

The paper demonstrates high quality in terms of methodology and experimental rigor. The authors provide a detailed and systematic evaluation pipeline, ensuring reproducibility and realistic assessments. The extensive experimentation and thorough documentation of the datasets and methods enhance the credibility of the findings.

The paper is well-organized and clearly written, with distinct sections that guide the reader through the background, methodology, datasets, experiments, and results.

The originality of the paper lies in the introduction of TGB 2.0, which addresses significant gaps in the evaluation of multi-relational temporal graphs.

The work is highly significant as it provides a standardized framework for evaluating methods on large-scale temporal graphs. This is crucial for advancing research in various applications, such as recommendation systems, knowledge base completion, and molecular learning. Besides, the insights gained from the experiments offer valuable directions for future research on scalable methods and the importance of leveraging multi-relational information.


### Pros:

- The paper provides valuable insights into the performance and scalability of various methods, highlighting the importance of edge-type information and the need for more scalable approaches.

- The introduction of eight novel datasets significantly enhances the scope and depth of temporal graph evaluations.

- The automated pipeline ensures reproducible and realistic evaluations, addressing previous inconsistencies in evaluation metrics.

- The code are easily approachable and ensures the reproducibility of the work.

### Cons:

- The framework's complexity may limit accessibility for some researchers, particularly those with limited computational resources.

**Strengths:**

- The introduction of TGB 2.0 addresses critical gaps in the evaluation of temporal graphs, providing a robust framework that can significantly advance the field.

- The standardized evaluation pipeline and large datasets are valuable resources for researchers in various domains, facilitating more comprehensive and comparative studies.

- The methodological rigor and extensive experimentation demonstrate a high level of research quality.

**Additional Feedback:**

Overall, the paper makes a significant contribution to the field by providing a robust framework for evaluating multi-relational temporal graphs.

I have one minor question:

Will the hyperparameters really affect the performance of each method a lot?

**Clarity:**

The paper is well-written, with clear explanations of complex concepts and methodologies. The use of detailed figures and tables helps illustrate the results effectively, enhancing the overall clarity.

**Correctness:**

The claims made in the submission are well-supported by the data and methods presented. The experimental design and evaluation methods are appropriate and performed correctly, ensuring the validity of the findings.

**Documentation:**

The documentation of the dataset and methods is adequately detailed and sufficient to support reproducibility. The paper includes information on data collection, organization, and intended uses, ensuring responsible and ethical use.

**Ethics:**

There are no significant ethical concerns with the submission. The study adheres to ethical standards for data privacy, consent, and responsible use, aiming to advance scientific knowledge and methodological innovation.

**Limitations:**

The authors have adequately addressed the limitations of their work, discussing the scalability challenges and the complexity of implementing the framework. They provide constructive suggestions for future research, focusing on more scalable methods and broader dataset validations. The potential negative societal impacts, particularly regarding data privacy and ethical considerations, are also discussed, with steps taken to mitigate these issues.

**Opportunities For Improvement:**

I am quite satisfied with the submission and could not detect any shortcomings. (Would be happy to discuss with other reviewers.)

**Relation To Prior Work:**

The paper provides a thorough discussion of how it builds on and differs from previous work, situating its contributions within the context of existing research on temporal graph learning. The integration of multi-relational temporal graphs into the benchmark represents a significant advancement.

**Summary And Contributions:**

The paper presents Temporal Graph Benchmark 2.0 (TGB 2.0), a comprehensive benchmarking framework tailored for evaluating methods predicting future links on Temporal Knowledge Graphs (TKGs) and Temporal Heterogeneous Graphs (THGs). TGB 2.0 extends the existing Temporal Graph Benchmark by introducing eight new datasets across five domains, significantly larger than existing datasets, facilitating robust and reproducible evaluations. The paper highlights key insights from extensive experiments, such as the importance of edge-type information for high performance, the competitive nature of simple heuristic baselines, and the scalability challenges of most methods on the largest datasets. The contributions include large and diverse datasets, a realistic evaluation pipeline, and valuable experimental insights.

---

> ### Author Rebuttal · Authors · 2024-08-16
>
> ## Response to Reviewer NhyZ
>
> We thank the reviewer for taking time to assess our paper and offering valuable suggestions. It's rewarding to see the recognition of key aspects of our work, e.g. that the pipeline ensures reproducible and realistic evaluations, approachability of our code, the new dataset to enhance scope and depth, and the experimental insights. In the following we will reply to the reviewers question and comment.
>
> > ***Q1 Will the hyperparameters really affect the performance of each method a lot?***
> >
> We thank the reviewer for raising this point. In the current work, we relied on the hyperparameter settings and findings reported in the original papers for each method. Below, we summarize the key insights provided by those studies to give an understanding of how hyperparameters may impact performance:
>
> * Tlogic:  The performance saturates between 100 and 200 walks; including rules of lengths 1,2, and 3 results in best performance, but simple rules of length 1 already yield very good performance while reducing computation time and memory usage; the performance increases with the size of the time window.
> * CEN and RE-GCN do not conduct hyperparameter studies.
> * Recurrency Baseline: The time decay λ significantly affects the MRR. This is also reflected in our results (see Table 2 in our paper), where the automatic selection of λ (Recurrency Baseline train) leads to higher performance compared to using the default value (Recurrency Baseline default), though the latter results in significantly reduced runtimes.
> * TGN: The presence of the memory module allows a single layer to obtain high performance. This is because when accessing the memory of the 1-hop neighbors, they are indirectly accessing information from hops further away. Moreover, being able to use only one layer of graph attention speeds up the model dramatically. However, a detailed study on the additional hyperparameters (e.g., embedding dimension, dropout) is not included.
> * STHN: Performance tends to slightly decline with longer interaction sequences.
> * Edgebank: heuristics methods, requires no hyperparameters.
>
>
> > ***W1 The framework's complexity may limit accessibility for some researchers, particularly those with limited computational resources.***
>
> We thank the reviewer for this feedback. Indeed, providing accessible datasets for researchers with limited computational resources is important. In the design of TGB 2.0, we actually have this in mind. We classify datasets into small, medium and large scales. Small datasets such as `tkgl-smallpedia` and `thgl-software` require comparatively low computational resources and are ideal for fast prototyping of research ideas. The medium size datasets are then designed to test the methods on larger and more challenging data. Lastly, the large size datasets are designed to be closest to the sizes used in practice and allows for testing the scalability of methods.
>
> In addition, to lower the complexity of using the TGB 2.0 framework, we provide a [pip package](https://pypi.org/project/py-tgb/) for easy installation as well as detailed [example scripts](https://github.com/shenyangHuang/TGB/tree/main/examples) for how to run baselines on each dataset. Moreover, documentations are provided on the [TGB website](https://tgb.complexdatalab.com/) as well as [code documentation website](https://docs.tgb.complexdatalab.com/)
>
> Lastly, TGB 2.0 is a community driven project, we plan to incorporate new datasets and suggestions from the research community. The open sourced code for TGB evaluation also enables researchers to use the same evaluation pipeline on their custom datasets.

---

> > ### Comment · Reviewer_NhyZ · 2024-08-16
> >
> > Dear Authors,
> >
> > Thank you for your detailed response. I am pleased to see that my concerns have been thoroughly addressed.
> >
> > Warm regards,
> >
> > Reviewer NhyZ

---

> > > ### Author Response · Authors · 2024-08-16
> > > **Thank you**
> > >
> > > Dear Reviewer NhyX
> > >
> > > Thank you for your reply!
> > > We are happy to have addressed your concerns.
> > >
> > > All the best

---

### Official Review · Reviewer_ADfV · 2024-07-21
**Review of Submission133**

**Rating:** 5
**Confidence:** 4
**Correctness:** Yes.
**Clarity:** Yes.

**Review:**

Pros：
1.	The paper is based on TGB's work and is clear and easy to understand.
2.	The charts and tables in the paper are relatively standardized.
3.	The code and dataset of the paper have been open-sourced. For the datasets used in this paper, the paper provides the source and open-source protocol.

Cons：
1.	This paper is based on the existing benchmark, TGB. However, in terms of innovation and novelty in the research field - Temporal Knowledge Graphs and Heterogeneous Graphs, it is relatively insufficient.

**Strengths:**

This paper aims to address two main challenges: inconsistent evaluation and limited dataset size in benchmarking on multi-relational temporal graphs. The strengths of this paper are as follows:
1. TGB 2.0: The paper presents Temporal Graph Benchmark 2.0, an advanced benchmarking framework aimed at evaluating machine learning methods for predicting future links in temporal graphs. This framework addresses the need for standardized and robust evaluations in this area of research.
2. Extensive Dataset Collection: TGB 2.0 includes eight new datasets that span five different domains, significantly enhancing the scale of datasets available for research with up to 53 million edges. These datasets surpass previous benchmarks in terms of the number of nodes, edges, and timestamps, providing a more challenging and realistic setup for model evaluation.
3. Reproducible Evaluation Pipeline: The benchmark introduces a reproducible and realistic evaluation pipeline that helps in assessing the performance of machine learning models on large-scale, complex temporal graphs. This pipeline is crucial for achieving consistent and reliable research outcomes.
4. Insights from Experimental Results: The paper provides valuable insights from extensive experiments using TGB 2.0. It finds that leveraging edge-type information is vital for high performance, simple heuristic baselines often perform comparably to more complex models, and most current methods struggle with the largest datasets, indicating a pressing need for research into more scalable solutions.

**Additional Feedback:**

N/A

**Documentation:**

Yes.

**Ethics:**

No.

**Limitations:**

Refer to Opportunities for Improvement.

**Opportunities For Improvement:**

1. TGB 2.0 aims to address two main challenges: inconsistent evaluation and limited dataset size. However, there are many similar research works, and the innovation and novelty of this paper are relatively insufficient.
2. There are many OOMs and OOTs in Tables 2 and 3, which means that how to train relatively large temporal graphs is the key to this field - Temporal Knowledge Graphs and Heterogeneous Graphs. The relatively large datasets collected in the paper are very meaningful. Therefore, the complete test results are highly anticipated.

**Relation To Prior Work:**

Yes.

**Summary And Contributions:**

The paper introduces the Temporal Graph Benchmark 2.0 (TGB 2.0). This benchmark framework is designed specifically to evaluate predictive models on temporal knowledge graphs and heterogeneous graphs, emphasizing large-scale datasets. TGB 2.0 improves upon its predecessor by including eight new datasets across five domains, featuring up to 53 million edges, which are larger in scale compared to previous datasets. It also provides a reproducible and realistic evaluation pipeline tailored for multi-relational temporal graphs.
Contributions:
1.	The paper presents Temporal Graph Benchmark 2.0, an advanced benchmarking framework aimed at evaluating machine learning methods for predicting future links in temporal graphs.
2.	TGB 2.0 includes eight new datasets that span five different domains, significantly enhancing the scale of datasets available for research with up to 53 million edges.
3.	The benchmark introduces a reproducible and realistic evaluation pipeline that helps in assessing the performance of machine learning models on large-scale, complex temporal graphs.
4.	The paper provides valuable insights from extensive experiments using TGB 2.0. It finds that leveraging edge-type information is vital for high performance, simple heuristic baselines often perform comparably to more complex models, and most current methods struggle with the largest datasets, indicating a pressing need for research into more scalable solutions.

---

> ### Author Rebuttal · Authors · 2024-08-16
>
> ## Response to Reviewer ADfV
>
> We thank the reviewer for taking time to assess our paper and offering valuable suggestions. We are happy that the reviewer describes our work as clear and easy to understand, and appreciates the charts, tables and open source code and datasets. In the following, we hope to address your concerns regarding innovation and novelty.
>
> > ***W1 This paper is based on the existing benchmark, TGB. However, in terms of innovation and novelty in the research field - Temporal Knowledge Graphs and Heterogeneous Graphs, it is relatively insufficient.***
>
> We thank the reviewer for this feedback, We would like to emphasize several important novel contributions of our work:
>
> 1. **Expanding to temporal** ***multi-relational*** **networks:** the original TGB datasets only focus on single-relation temporal graphs and lack any multi-relational networks (i.e. no datasets with multiple edge types). This work significantly expands the scope of the original TGB by including novel datasets for both temporal knowledge graphs and temporal heterogeneous graphs, both of which forms a dedicated subfield within temporal graph learning due to their complexity. Therefore, TGB 2.0 addresses the need for well-curated TKG and THG datasets.
> 2. **Novel, diverse and large scale datasets:** In this work, we include a total of eight novel datasets across five diverse domains, from small sized datasets for prototyping research methods to large scale datasets matching the size of networks in applications. The newly introduced TKG datasets are orders of magnitude larger than existing ones while the THG datasets are also significantly larger than available ones.
> 3. **Standardized Procedures and Negative Sampling:** We also propose the first  standardized evaluation framework for multi-relational graphs. Further, we introduce novel  negative sampling strategies, specifically designed for temporal multi-relational graphs, such as the *1-vs-q* strategy, based on the destination nodes of each edge type. Empirically we verified that these negative sampling strategies achieve a good tradeoff between efficiency and evaluation completeness.
> 4. **Differences in Baselines:** We also evaluate a comprehensive set of baseline models that are specifically designed or adapted for multi-relational temporal graphs, meaning the baselines are able to benefit from the additional edge type information.
> 5. **Open and Reproducible Framework:** We open source our code base on github (also available as a pip package)  as well as provide public and permanent access to TGB 2.0 datasets.
>
> We hope this clarification addresses your concerns regarding the innovation and novelty of our work. We believe that our contributions significantly advance the state of research in TKGs and THGs and offer existing opportunities for future exploration.
>
> > ***There are many OOMs and OOTs in Tables 2 and 3, which means that how to train relatively large temporal graphs is the key to this field - Temporal Knowledge Graphs and Heterogeneous Graphs. The relatively large datasets collected in the paper are very meaningful. Therefore, the complete test results are highly anticipated.***
>
> Thank you for recognizing the significance of the large datasets we've collected. We agree that the presence of numerous OOM and OOT instances in Tables 2 and 3 highlights a critical challenge in the field of Temporal Knowledge Graphs and Heterogeneous Graphs—namely, the difficulty of training on relatively large multi-relational temporal graphs. This underscores the importance of future work focused on improving the scalability of current methods, or the introduction of more scalable approaches.
> For our experiments, we use the largest GPU available to us with 40 GB of memory. However, due to the significantly larger size of our novel datasets, OOM errors and OOT errors are still encountered. We discuss the computation resources in detail in Appendix G.1.
> As you pointed out, generating results for these large-scale datasets remains a significant hurdle, with existing methods often unable to handle them effectively. While addressing scalability is indeed a crucial area for future research, it was beyond the scope of our current work to optimize existing methods for this purpose.

---

> > ### Author Response · Authors · 2024-08-24
> > **Official Comment by Authors**
> >
> > Thank you again for your insight review and feedback. As we are approaching the end of the rebuttal period, if you have any further questions, we would be glad to answer them.

---

### Official Review · Reviewer_pUeu · 2024-07-25
**Review for "TGB 2.0: A Benchmark for Learning on Temporal Knowledge Graphs and Heterogeneous Graphs"**

**Rating:** 7
**Confidence:** 3

**Review:**

The quality of the submission is high, presenting a clear and detailed benchmark that addresses a significant need in the study of temporal graphs. The originality lies in its focus on large-scale, multi-relational temporal graphs and the comprehensiveness of the evaluation. The significance is underscored by its potential to facilitate more effective and scalable graph modeling techniques.

Pros:

1. Addresses a critical gap in available resources for temporal graph analysis.

2. Provides a clear, rigorous introduction and statistical analysis of new, large-scale datasets.

3. Demonstrates the utility of these datasets across various domains, enhancing their relevance and applicability.

Cons:

1. Lacks detailed discussion on the dataset construction process and the standardization of data formats.

2. Limited experimental analysis that could benefit from additional metrics and broader methodological comparisons.

3. Focuses primarily on link prediction, missing the opportunity to explore other graph-related tasks.

**Strengths:**

The paper is significant due to its practical relevance to applications that require temporal and heterogeneous graph analysis. The quality of the research is evident in the meticulous dataset preparation and the thorough analysis provided. The submission is also socially beneficial, as better benchmarks can lead to advancements in technologies that rely on dynamic graph data.

**Additional Feedback:**

none

**Clarity:**

The paper is generally well-written, with a clear structure and a logical flow. However, some sections could be expanded for clarity, particularly those discussing dataset construction and the specifics of the experimental setup.

**Correctness:**

The claims regarding the benchmark's capability and the datasets' scale and diversity appear correct and are supported by the experimental results. The benchmark construction and evaluation methods are sound and appropriate for the stated goals.

**Documentation:**

The documentation of the datasets and benchmark framework is good, but could be improved by providing more detailed information on data collection, organization, and the protocols used for ensuring the datasets' long-term maintenance and ethical use.

**Ethics:**

no ethic problem

**Limitations:**

The discussion on limitations is somewhat cursory. The paper would benefit from a deeper examination of the scalability of the methods tested, particularly on the largest datasets. Additionally, the potential negative impacts, such as privacy concerns related to temporal data, are not addressed.

**Opportunities For Improvement:**

The paper could improve by expanding the discussion on dataset construction and including more comprehensive experimental evaluations. Specifically, details on data cleaning, format standardization, and the construction pipeline would enhance reproducibility and trust in the benchmark's reliability.

**Relation To Prior Work:**

The submission adequately discusses its advancements over prior benchmarks and datasets, clearly highlighting its contributions and improvements in scale, diversity, and realism.

**Summary And Contributions:**

The submission introduces Temporal Graph Benchmark 2.0 (TGB 2.0), a framework designed for evaluating machine learning models on Temporal Knowledge Graphs (TKG) and Temporal Heterogeneous Graphs (THG). It extends previous benchmarks by providing larger datasets across multiple domains and implementing a reproducible evaluation pipeline. Key contributions include the creation of eight diverse and large-scale datasets and insights into the performance of various predictive methods on these graphs.

---

> ### Author Rebuttal · Authors · 2024-08-16
>
> ## Response to Reviewer pUeu
> We thank the reviewer for taking time to assess our paper and offering valuable suggestions. We are pleased that the reviewer recognizes the importance of our work in addressing a critical gap in available resources. Here, we address each of the points raised by the reviewer.
>
> > ***W1 Lacks detailed discussion on the dataset construction process and the standardization of data formats.***
>
> Thank you for raising this point. In addition to the dataset details presented in Section 4 of the original submission, we have added a new section on data processing details as well as more descriptions on the file types we provide for each dataset (see rebuttal pdf). We open source all data collection details and our data processing scripts can be found on Github.
>
>
> > ***W2 Limited experimental analysis that could benefit from additional metrics and broader methodological comparisons.***
>
> Thank you for raising this point. We decided to use the  MRR metric in the experiments as it is one of the most widely used evaluation metrics for link level tasks. We added a table of additional results for `tkgl-smallpedia`, `tkgl-polecat`, `tkgl-wikidata` datasets with the Hits@10 metric in the rebuttal pdf. Due to the time constraint, we aim to provide a full list of accompanying results in Hits@10 metric in the paper revision. The results here show that the ranking of methods in Hits@10 and MRR metric are closely matched, especially the top ranking methods are the same for both metrics.
>
> The baseline experiments in TGB 2.0 are a starting point for the leaderboard. TGB 2.0 leaderboards are open to community submissions and newly submitted methods will be constantly added to the leaderboard. We will incorporate newer metrics and suggestions from the community as the research on temporal graphs evolves.
>
> > ***W3 Focuses primarily on link prediction, missing the opportunity to explore other graph-related tasks.***
>
> Thank you for this comment. Link prediction is the most widely studied task on multi-relational temporal graphs, thus also having a large number of prior work designed for this task. Currently the tasks and datasets on TGB 2.0 provide a starting point for the community to explore tasks and scalability challenges on TKGs and THGs. We agree that additional tasks at node or graph level is an important future direction. Similarly to how TGB 2.0 significantly expands upon the original TGB work by adding new datasets and tasks for TKGs and THGs, we plan to continue to expand this work based on community feedback and newly available data sources. Due to the time constraint during the rebuttal period, we leave that as future work. One possible future direction is to incorporate recent work on graph level tasks such as the temporal graph property prediction task [1].
>
> [1] Shamsi K, Poursafaei F, Huang S, Ngo BT, Coskunuzer B, Akcora CG. GraphPulse: Topological representations for temporal graph property prediction. ICLR 2024.
>
> > ***C1 some sections could be expanded for clarity, particularly those discussing dataset construction and the specifics of the experimental setup.***
>
> Thank you for raising this point. We added dataset construction, cleaning, and format details in a new section (see rebuttal pdf). We open sourced all the data collection code on Github. Further, we provide more details on the evaluation procedure as well as open source the negative sample generation code on Github. Thus, researchers can use the same negative sample generation procedures for their own custom datasets. Lastly, all the baseline code are uploaded on Github. With this, we hope to ensure reproducibility and enhance our benchmark’s reliability. Further, we added details on the evaluation protocol in the appendix of the paper revision (see rebuttal pdf).
>
> > ***C2 The documentation [...] could be improved by providing more detailed information on data collection, organization, and the protocols used for ensuring the datasets' long-term maintenance and ethical use.***
>
> We thank the reviewer for this comment. In addition to the dataset collection details mentioned above, we discussed the dataset hosting information in Appendix D under the maintenance plan section. TGB 2.0 datasets are hosted via Digital Research Alliance of Canada funded by the Government of Canada for fast download speeds. We also made the datasets available on public platforms such as hugging face and Zenodo for permanent and public access.
>
> > ***C3 The discussion on limitations is somewhat cursory. The paper would benefit from a deeper examination of the scalability of the methods tested, particularly on the largest datasets. Additionally, the potential negative impacts, such as privacy concerns related to temporal data, are not addressed.***
>
> We thank the reviewer for raising this point. Indeed, scalability is an important future direction for the field of TKG and THG. In Appendix G.4 we provide details on the scalability challenges faced by different methods examined in the paper. Due to the constraint on compute resources available to us, we can only test on up to GPUs with 40 GB of memory. With the availability of TGB 2.0 datasets, we hope future scalable methods can be developed based on our datasets and provide additional empirical insight into how to efficiently learn from large TKG and THG data.
>
> Regarding potential negative impacts, we thank the reviewers for this comment and will add the following to the paper revision:
>
> ```
> As temporal graph data often records interactions between individuals or other sensitive information, data privacy is a potential concern. Sensitive information may be stored as node level or link level attributes thus proper anonymization of data should always be a top priority. In this work, we anonymize all user information where appropriate to prevent the leakage of identifiable information. For example, we anonymized user identifiers into integer IDs.
> ```

---

> > ### Comment · Reviewer_pUeu · 2024-08-20
> >
> > Thank you for your detailed response. My concerns have been fully addressed. I encourage you to incorporate the suggestions into the revised version of your paper.

---

> > > ### Author Response · Authors · 2024-08-20
> > >
> > > Dear Reviewer pUeu
> > >
> > > We thank you for your reply, and we are happy to have addressed your concerns.
> > > We thank you again for your suggestions to further improve the paper and will include them in the revised version.
> > >
> > > All the best

---

### Official Review · Reviewer_AdXk · 2024-07-26
**Accept**

**Rating:** 7
**Confidence:** 4
**Correctness:** NA
**Clarity:** NA

**Review:**

NA

**Strengths:**

1. Nice presentation with clear organization and nice figures

2. Complete experiments.

3. The paper is easy to follow with clear motivation

**Additional Feedback:**

NA

**Documentation:**

NA

**Opportunities For Improvement:**

Some important related works should be discussed.

[1] Cai, Borui, et al. "Temporal knowledge graph completion: A survey." arXiv preprint arXiv:2201.08236 (2022).

[2] Cai, Li, et al. "A Survey on Temporal Knowledge Graph: Representation Learning and Applications." arXiv preprint arXiv:2403.04782 (2024).

[3] Liang, Ke, et al. "A survey of knowledge graph reasoning on graph types: Static, dynamic, and multi-modal." IEEE Transactions on Pattern Analysis and Machine Intelligence (2024).

[4] Wang, Jiapu, et al. "A survey on temporal knowledge graph completion: Taxonomy, progress, and prospects." arXiv preprint arXiv:2308.02457 (2023).

**Relation To Prior Work:**

NA

**Summary And Contributions:**

The Temporal Graph Benchmark 2.0 (TGB 2.0) offers a robust framework for evaluating multi-relational temporal graph models, addressing the scarcity of standardized datasets and reproducibility issues. With eight new large-scale datasets, TGB 2.0 enhances comprehensive assessment capabilities. Findings from extensive tests emphasize the importance of edge-type information, the competitiveness of simple heuristics, and the need for more scalable methods, as most models cannot handle the largest datasets.

---

> ### Author Rebuttal · Authors · 2024-08-16
>
> ## Response to Reviewer AdXk
>
> We thank the reviewer for taking time to assess our paper and offering valuable suggestions on important related work. We are glad to receive positive feedback on our work from the reviewer. Following your feedback, we will updated our paper accordingly.
>
> > ***W1 Some important related works should be discussed.***
>
> We thank the reviewer for pointing out these references. We will add them to the revised version of the paper in Section 3 related work as follows:
>
> ```
> For more comprehensive discussions on methods and applications of TKG, we refer to the surveys [1], [2], [3], and [4].
> ```

---

> > ### Author Response · Authors · 2024-08-24
> > **Official Comment by Author**
> >
> > Thank you again for your insight review and feedback. As the end of the rebuttal period is approaching, if you have any further questions, we would be glad to answer them.

---

### Author Rebuttal · Authors · 2024-08-16

# Author Response to All Reviewers

We thank the reviewers for their insightful reviews and positive reception of our work.
We are glad to see that all reviewers found our paper to be clearly organized and easy to follow. Reviewers (pUeu and NhyZ) mentioned that our work is addressing a significant need and critical gap in available resources. Reviewers (pUeu, NhyZ and ADfV) acknowledged the diversity of TGB2’s datasets as well as its large scale and practical utility. Reviewer AdXk and NhyZ appreciate the rigor of experiment and Reviewer ADfV, NhyZ and pUeu agreed on the significance of insight from our empirical results.

We appreciate the constructive feedback from reviewers to help improve our paper. Here we upload a one page pdf which includes additional information that was requested by reviewers (new section on data processing details and additional results with Hits@k metric), We will also incorporate these suggestions into future revised versions of the paper.

TGB 2.0 is a community driven project and we plan to continue to improve the benchmark based on feedback from the reviewers and the community. Further, we address the comments of each reviewer individually below.

---

### Decision · Program_Chairs · 2024-09-26

**Decision:**

Accept (Poster)

**Comment:**

This work enhances the Temporal Graph Benchmark (TGB) framework with a focus on multi-relational temporal graphs (such as knowledge graphs and heterogeneous graphs) and a reproducible and realistic evaluation pipeline.

The reviewers unanimously acknowledge the high quality of the paper. Although opinions are mixed regarding the significance of the improvements over the previous TGB version, the overall sentiment leans positive, especially after the rebuttal and discussion period, which the meta-reviewer agrees with. Several additional concerns were raised, and the authors addressed most of them within the scope of their work.

Considering all factors, the meta-reviewer recommends accepting the paper